# Learning Unlabeled Clients Divergence for Federated Semi-Supervised Learning via Anchor Model Aggregation

**Marawan Elbatel**                                                      *mkfmelbatel@ust.hk*
*The Hong Kong University of Science and Technology , Hong Kong SAR, China*

**Hualiang Wang**                                                       *hwangfd@ust.hk*
*The Hong Kong University of Science and Technology , Hong Kong SAR, China*

**Jixiang Chen**                                                        *jchenhu@ust.hk*
*The Hong Kong University of Science and Technology , Hong Kong SAR, China*

**Hao Wang**                                                           *hw488@cs.rutgers.edu*
*Rutgers University, New Jersey, USA*

**Xiaomeng Li**                                                        *eexmli@ust.hk*
*The Hong Kong University of Science and Technology , Hong Kong SAR, China*

**Reviewed on OpenReview:** *https://openreview.net/forum?id=GDn6z9LIDs*

## Abstract

Federated semi-supervised learning (FedSemi) refers to scenarios where there may be clients with fully labeled data, clients with partially labeled, and even fully unlabeled clients while preserving data privacy. However, challenges arise from client drift due to undefined heterogeneous class distributions and erroneous pseudo-labels. Existing FedSemi methods typically fail to aggregate models from unlabeled clients due to their inherent unreliability, thus overlooking unique information from their heterogeneous data distribution, leading to sub-optimal results. In this paper, we enable unlabeled client aggregation through **SemiAnAgg**, a novel **Semi**-supervised **An**chor-Based federated **Agg**regation. SemiAnAgg learns unlabeled client contributions via an anchor model, effectively harnessing their informative value. Our key idea is that by feeding local client data to the same global model and the same consistently initialized anchor model (*i.e.*, random model), we can measure the importance of each unlabeled client accordingly. Extensive experiments demonstrate that SemiAnAgg achieves new state-of-the-art results on four widely used FedSemi benchmarks, leading to substantial performance improvements: a **9%** increase in accuracy on CIFAR-100 and a **7.6%** improvement in recall on the medical dataset ISIC-18, compared with prior state-of-the-art. Code is available at: https://github.com/xmed-lab/SemiAnAgg.

## 1 Introduction

Federated learning has emerged as a promising solution for learning in decentralized environments, where data centralization is often infeasible due to privacy concerns. Federated learning has gained considerable attention in machine learning tasks in natural image domains (McMahan et al., 2017) as well as medical image domains (Liu et al., 2021; Saha et al., 2023; Jiang et al., 2023). Due to the challenges of data and label heterogeneity, multiple methods such as MOON (Li et al., 2021a), FedDisco (Ye et al., 2023), FedFed (Yang et al., 2023), and FedRoD (Chen & Chao, 2022) have been developed, utilizing FedAvg (McMahan et al., 2017) as their baseline. While these methods have shown promise, they often assume that all clients have exhaustive and expert-level annotations, which is a requirement that is both time-consuming and labor-intensive. This renders them impractical in real-world cross-silo federated settings, such as those found in hospitals. For

instance, in a federated learning framework where multiple hospitals collaborate to develop a shared model for complex tasks like lesion classification, some hospitals may provide fully labeled data for training, while others, with limited expert manpower, can only offer unlabeled or partially labeled data for model training. Therefore, developing methods that require minimal expert-level annotations in decentralized settings is a crucial area of research that deserves further attention.

Federated Semi-Supervised Learning (FedSemi) arises to alleviate exhaustive expert-level annotations by utilizing unlabeled data along with labeled data to benefit the global performance in many recent works (Zhang et al., 2023; Cho et al., 2023; Zhang et al., 2023; Wang et al., 2020; Jeong et al., 2021; Lin et al., 2021; Liu et al., 2021; Saha et al., 2023; Liang et al., 2022; Li et al., 2023; Kim et al., 2022; 2023b). Researchers have explored various settings for FedSemi, with the most generalizable setting considering fully unlabeled clients and other clients that can be either labeled or partially labeled, in either independent and identically distributed (IID) or non-IID settings (Liang et al., 2022; Li et al., 2023).

Despite the effectiveness of these methods, existing approaches have two crucial limitations. First, their aggregation typically follows the standard FedAvg (McMahan et al., 2017), which aggregates clients based on the volume of local data, failing to account for the fact that labeled clients can generate more accurate information than unlabeled clients, regardless of data volume. For example, when the number of unlabeled clients is dominant, the global model becomes heavily influenced by the unlabeled clients, leading to limited performance primarily due to errors introduced by the pseudo-labels assigned to the unlabeled data. Second, existing methods treat the divergence of unlabeled clients, calculated in the gradient space, as noise due to the unreliability of the pseudo-labels used in training. It is worth mentioning that client divergence may reflect the presence of minority classes and unique attributes within the dataset, rather than mere noise.

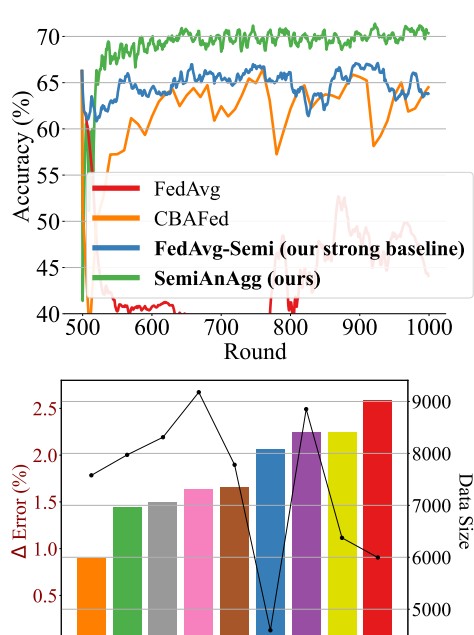

To address the first limitation, we first show that aggregation based on the local dataset size, as in FedAvg (McMahan et al., 2017), falls short in the FedSemi setting. At first glance, this may not be surprising: an unlabeled client with a large amount of data but incorrect pseudo-labels should not dominate the global optimization. Recall that traditional semi-supervised learning (SSL) typically seeks a balance between optimizing the empirical risks associated with both labeled and unlabeled data to avoid skewing toward potentially incorrect pseudo-labels (Tarvainen & Valpola, 2017; Sohn et al., 2020; Zhang et al., 2021; Chen et al., 2023; Wang et al., 2023). Therefore, by balancing the empirical risk of labeled and unlabeled clients during global model aggregation, we present a stronger baseline for FedSemi termed **FedAvg-Semi**. Figure 1 shows that our simple baseline, **FedAvg-Semi**, can achieve **comparable performance** to **SOTA of FedSemi**, CBAFed (Li et al., 2023). Nevertheless, treating unlabeled clients based on their local dataset size is suboptimal: an unlabeled client with a small dataset size may contain underrepresented attributes and minority classes that are not present in others.

Figure 1: **Upper:** FedAvg (McMahan et al., 2017) fails miserably. Yet, disentangling the aggregation (FedAvg-Semi) achieves performance comparable to the state-of-the-art FedSemi method, CBAFed (Li et al., 2023). By promoting diversity among unlabeled clients our **SemiAnAgg** achieves a new SOTA on four FedSemi benchmarks. **Lower:** Leave one unlabeled client out. The bar refers to the Δ Error (%), and the line refers to the Data Size. In the leave-one-out experiment setting, one unlabeled client is excluded during training. The results indicate that the decrease in accuracy (represented by the bar) does not correspond proportionally with the data size (represented by the line).

The second limitation of current methodologies is their failure to account for the diverse contributions of unlabeled clients, particularly in reflecting the presence of minority classes and unique attributes within majority classes. To validate the importance of unlabeled clients' diverse contributions, we conducted an extensive empirical evaluation using the popular *leave-one-out* valuation strategy (Ghorbani & Zou, 2019). As shown in Figure 1, the leave-one-out valuation strategy reveals significant differences in

how unlabeled clients contribute to the overall performance of FedSemi, allowing us to draw two important conclusions. First, there is no correlation between the size of the unlabeled client's dataset and the value a client contributes to the system. For instance, excluding client 9 significantly degrades performance ($\Delta$ Error $\uparrow$) compared to when client 2 or 3 are removed, even though client 9 have lower amount of data. Second, the varying impact of removing different unlabeled clients showcases the importance of each unlabeled client accordingly. Therefore, measuring the value of unlabeled clients irrespective of their data size is a challenge yet to be tackled in FedSemi.

To this end, we propose a novel aggregation, **SemiAnAgg**, which learns the importance of unlabeled clients via a consistently randomly initialized model. Our key idea is that by feeding the local client data to the same global model and the same consistently initialized random anchor model, we can measure the importance of each unlabeled client. Specifically, our intuition is two-fold: 1) The optimal feature representation for client data should differ from a random representation, as random representations lack meaningful structure. Therefore, clients contributing to representations deviating from randomness are more likely to steer the model toward the global optimum. 2) Since all clients share the same global and anchor model, the inter-client data variability can be measured consistently. In other words, the expectation of the global model on client data that is more distant indicates a more diverse distribution covering unique information. Consequently, SemiAnAgg gives higher weights to these clients, harnessing their potential. Without requiring any direct sharing of data, features, or gradient information among the clients and maintaining the same communication costs as previous FedSemi approaches with minimal computation and storage overhead, **SemiAnAgg** achieves state-of-the-art results on four widely used benchmarks with improvements up to **9%** in accuracy on CIFAR-100, **9.5%** on the imbalanced CIFAR-100LT, and a **7.65%** improvement in recall on the medical dataset ISIC-18.

Our contributions can be summarized as follows:

- We provide a new insight for FedSemi that was neglected in prior work: measuring the importance of unlabeled clients.
- Unlike previous FedSemi approaches that treat the divergence of unlabeled clients as noise, we propose **SemiAnAgg**, a novel aggregation method that effectively aggregates the most informative unlabeled clients, significantly harnessing their unique information.
- Our method consistently outperforms prior state-of-the-art methods on four widely used benchmarks, surpassing SOTA CBAFed by 9% in accuracy on CIFAR-100, 9.5% on its highly imbalanced version, CIFAR-100LT, and achieving a 7.65% recall improvement on the medical dataset, ISIC-18.

## 2 Related Work

### 2.1 Semi-Supervised Learning

**Semi-Supervised Learning (SSL)** leverages a large amount of unlabeled data along with labeled data to learn a generalizable model. Semi-supervised methods include consistency regularization (ensuring consistency between two distorted unlabeled images) (Li et al., 2021b; Tarvainen & Valpola, 2017), generating pseudo labels through supervised objectives (Zhang et al., 2021; Chen et al., 2023; Wang et al., 2023), or self-supervised clustering objectives (Fini et al., 2023). Prominent methods incorporating adaptive pseudo labeling are FlexMatch (Zhang et al., 2021), SoftMatch (Chen et al., 2023), and FreeMatch (Wang et al., 2023). However, these strategies assume uniform and identical class distribution of labeled and unlabeled data, which is often not the case in real-world applications.

**Semi-Supervised Imbalanced Learning (SSIL)** aims to learn SSL models in scenarios where there is a class imbalance distribution in labeled and unlabeled data. Several works address this issue by amplifying pseudo labels for minority classes through resampling (Wei et al., 2021), re-weighting (Wei et al., 2022; Wang & Li, 2023a;b), and classifier blending (Oh et al., 2022). However, these methods assume the unlabeled class imbalance distributions follow the labeled ones. In the non-IID FedSemi setting, a more realistic scenario arises where there is a class distribution mismatch between labeled and unlabeled data. This setting, which has not been extensively explored, is addressed by the state-of-the-art method proposed by ACR (Wei & Gan, 2023). ACR builds upon FixMatch (Sohn et al., 2020) with a dual branch and introduces adaptive

consistency regularization. The intensity of logit adjustment, controlled by a scaling parameter, is adaptively calculated based on the pseudo-label distribution distance to three anchor distributions: uniform, labeled distribution, and its reversed version. While ACR (Wei & Gan, 2023) showed promising results in centralized settings, the three anchor distribution is crafted and does not adhere to non-IID FedSemi. Thus, the challenge of semi-supervised learning with heterogeneous labeled and unlabeled class distributions, particularly in FL non-IID scenarios, remains unresolved.

## 2.2 Federated Semi-Supervised Learning

Federated Semi-Supervised Learning (FedSemi) addresses the decentralized learning of both unlabeled and labeled data while preserving privacy. FedSemi has been explored in three different settings. In the first two settings, labeled data is available on the server, as in FedMatch (Jeong et al., 2021), Semi-FL (Diao et al., 2022), and FedLID (Psaltis et al., 2023), or clients have partially labeled data, as seen in SemiFed (Lin et al., 2021) and FedLabel (Cho et al., 2023). The third setting, which we consider more realistic, involves fully unlabeled clients, while other clients can be either fully labeled or partially labeled. For the third setting, recent works have shifted from addressing clients with independent and identically distributed (IID) data, such as FedConsist (Wang et al., 2020) and FedIRM (Liu et al., 2021), to non-IID data, including RSCFed (Liang et al., 2022), IsoFed (Saha et al., 2023), and CBAFed (Li et al., 2023). This work focuses on the third setting, characterized by most clients with fully unlabeled and potentially non-IID data.

## 2.3 Federated Model Aggregation

Federated aggregation aims to improve the global model through proper aggregation across various clients. Given the heterogeneity of client data, clients with unique information could benefit the global model more during optimization, rendering the control of clients' contributions a crucial concern. This direction has shown great promise in supervised federated learning settings (Jiang et al., 2023; Elbatel et al., 2023; Ye et al., 2023). For instance, FedCE (Jiang et al., 2023) measures contribution in both the gradient and sample space. Using the sample space involves prediction error calculation, relying on labeled validation data within clients. FedMAS (Elbatel et al., 2023) proposes leveraging labeled data for estimating inter-client intra-class variation to emphasize the contribution of clients with minority classes. However, these methods rely on label information, making them inapplicable to unlabeled clients in FedSemi. In unlabeled clients, the non-IID distribution results in noisy pseudo-label estimation, rendering the surrogation with pseudo-labels on unlabeled clients less effective for aggregation measurement.

Existing FedSemi (Liang et al., 2022; Li et al., 2023) methods regard non-robust clients or data as outliers and opt to minimize their influence. RSCFed (Liang et al., 2022) proposes to leverage unlabeled gradient divergence to eliminate non-robust (noisy) clients through average consensus inspired by RANSAC (Fischler & Bolles, 1981). However, RSCFed (Liang et al., 2022) does not consider cases when unlabeled clients are diverse due to unique information (i.e. underrepresented classes or attributes). IsoFed (Saha et al., 2023) proposes alternating model training between labeled and unlabeled clients in each round, which might result in significant catastrophic forgetting after each round. CBAFed (Li et al., 2023) introduced federated class adaptive pseudo labeling, which can be seen as curriculum-paced pseudo learning from a global perspective (Zhang et al., 2021). It uses hard temporal ensembling (Rasmus et al., 2015; Tarvainen & Valpola, 2017; Laine & Aila, 2017) to address the stochastic variability in the global model and simply aggregates models based on reliable data amount. These strategies overlook the informative unlabeled clients with heterogeneous data, leading to suboptimal optimization. Therefore, a more comprehensive method for quantifying unlabeled clients is needed. In this work, we fill this gap by introducing SemiAnAgg, an anchor-based aggregation strategy in FedSemi, which harnesses the heterogeneity of unlabeled clients during valuation.

# 3 Preliminaries on Federated Semi-Supervised Learning

**FedSemi Setting.** Let us consider the generic Federated Semi-Supervised Learning (FedSemi) setting, which allows clients to be fully unlabeled, partially labeled, or fully labeled which is introduced

in RSCFed (Liang et al., 2022), IsoFed (Saha et al., 2023), and CBAFed (Li et al., 2023). We denote the set of clients as $\{\mathcal{C}_1, ..., \mathcal{C}_K\}$, where each client $\mathcal{C}_k$ possesses a local dataset represented by $\mathcal{D}_{client} = \left\{ \left\{ \left( X_i^L, y_i^L \right) \right\}_{i=1}^{N^L}, \left\{ \left( X_i^U \right) \right\}_{i=1}^{N^U} \right\}$. Here, $N^L$ and $N^U$ correspond to the number of labeled and unlabeled data instances, respectively. The objective is to derive a robust global model, $\theta_{glob}$, which effectively leverages all the available data across all clients. The most rigorous setting, as defined in (Liang et al., 2022; Li et al., 2023), is characterized by the majority of clients possessing fully unlabeled non-IID data, with $N^L = 0$.

**Federated Warm-up.** To ensure reliable pseudo-labels for unlabeled clients in initial stages, previous FedSemi approaches (Li et al., 2023; Liang et al., 2022) have employed a warm-up phase based on labeled clients using traditional FedAvg (McMahan et al., 2017). In this paper, we follow the generic FedSemi benchmark (Liang et al., 2022; Li et al., 2023) in label-dependent federated warmup for all reported results, yet we present additional experiments in Appendix A with federated self-supervised warmup (Zhang et al., 2020; Lubana et al., 2022; Zhuang et al., 2022; Kim et al., 2023a) highlighting their effectiveness in case of working with limited and highly imbalanced datasets (Yang & Xu, 2020; Yan et al., 2022).

**Local Training.** Each client $\mathcal{C}$ performs $E$ local steps based on the local dataset $\mathcal{D}_{client}$. Given our proposed *aggregation method is perpendicular to self-training regimes* (Tarvainen & Valpola, 2017; Chen et al., 2023; Wang et al., 2023), we adopt the canonical supervised training and self-training regimes on labeled and unlabeled data, respectively. Specifically, we utilize FlexMatch (Zhang et al., 2021) as our baseline to minimize a local objective function:

$$\mathcal{L} = \lambda_1 \mathcal{L}_{sup} + \lambda_2 \mathcal{L}_{unsup}, \tag{1}$$

where $\lambda_1$ is equal to zero in fully unlabeled clients, and $\lambda_2$ is the pre-defined weight coefficient between supervised and unsupervised loss terms. We fix $\lambda_2$ to 1 as traditional pseudo labelling approaches (Sohn et al., 2020; Wang et al., 2023; Zhang et al., 2021).

## 4 Method

### 4.1 FedAvg-Semi: A Strong Aggregation Baseline for Fedsemi

Previous FedSemi approaches (Cho et al., 2023; Liang et al., 2022; Li et al., 2023) show that assigning a higher weight to labeled clients on the server produces better performance than traditional FedAvg aggregation (McMahan et al., 2017). Notably, semi-supervised learning necessitates a weighted combination of supervised and unsupervised loss (Equation 1), with prior semi-supervised regimes (Sohn et al., 2020; Wang et al., 2023; Zhang et al., 2021) equally weighting $\mathcal{L}_{sup}$ and $\mathcal{L}_{unsup}$ to avoid bias towards potentially incorrect pseudo-labels. Consequently, it is crucial to **re-adjust the global optimization objective in federated semi-supervised learning**. Unlike existing FedSemi approaches (Cho et al., 2023; Liang et al., 2022; Li et al., 2023) that aggregate clients on the server based on traditional FedAvg (McMahan et al., 2017), we disentangle the aggregation based on labeled and unlabeled data on the server to write a more generic form, termed FedAvg-Semi.

Given $K$ clients, with number of labeled samples, $N^L$, and the selected number of unlabeled samples contributing for $\mathcal{L}_{unsup}$ in each client local training, $\hat{N}^U$, FedAvg-Semi dynamically disentangles FedAvg (McMahan et al., 2017) to ensure a global optimization objective consistent with traditional semi-supervised optimization as follows:

$$\theta_{global} = \sum_{k=1}^{K} \hat{\lambda}_1 \underbrace{\left( \frac{N_k^L}{N_{total}^L} \right)}_{\text{Supervised}} \theta_k + \hat{\lambda}_2 \underbrace{\left( \frac{\hat{N}_k^U}{\hat{N}_{total}^U} \right)}_{\text{Unsupervised}} \theta_k, \ N_{total}^L = \sum_{k=1}^{K} N_k^L \ , \text{ and } \hat{N}_{total}^U = \sum_{k=1}^{K} \hat{N}_k^U, \tag{2}$$

where $\hat{\lambda}_1 + \hat{\lambda}_2 = 1$ to ensure a normalized mean, and control the optimization of the labeled and unlabeled data consistent with semi-supervised regimes. Setting $\hat{\lambda}_1 = 1$ reduces to supervised warm-up on labeled clients. For simplicity, we set $\hat{\lambda}_1 = \hat{\lambda}_2 = 0.5$, noting that ideally these values could be ramped during federation.

## 4.2 SemiAnAgg: Semi-Supervised Anchor-Based Aggregation

Unlabeled clients in semi-supervised learning can provide valuable and diverse information, but their local models are unreliable due to erroneous pseudo-labels. In FedSemi, aggregation strategies typically aim to identify and treat unlabeled clients' local gradient divergence as noise. Nevertheless, this approach overlooks that such divergence could be due to informative, underrepresented attributes and classes in the data. Therefore, it is crucial to develop aggregation strategies that can leverage the full potential of unlabeled clients' data.

To this end, we propose a novel aggregation strategy to re-think the valuation of unlabeled clients in FedSemi, capturing their diversion without relying on labeled samples. Specifically, we propose to **measure** the **diversity** in the feature space based on a **reference embedding** from a **consistently initialized anchor model** across clients.

At the start of the federated process, given the locally unlabeled samples: $\left\{\left(X_i^U\right)\right\}_{i=1}^{N^U}$, we store a dictionary of feature representations as: $\{(q_i)\}_{i=1}^{N^U}$, where $q_i = \theta_{anch}^{enc}(X_i)$, and $\theta_{anch}^{enc}$ represents a Kaiming initialized **random encoder with the same seed across clients** (He et al., 2015). The motivation for using random encoders stems from the realm of generative model evaluation, where they have been shown to provide macroscopic views on distributional discrepancies (Naeem et al., 2020). The storage footprint of the feature dictionary varies, from 968KB for a client with 484 samples to 5.01MB for a client with 2567 samples, totaling 15.6MB across all clients for the ISIC-18 dataset-**a mere 0.006 of the dataset's 2.5GB size.**

During the federated process, the client receives the averaged global model $\theta_{glob}$ from the server to update $\theta_{local}$. From this, we can extract the feature

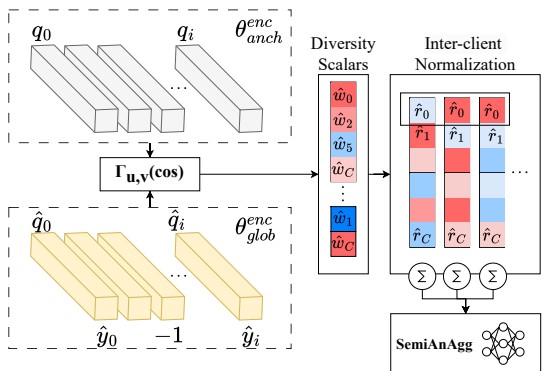

Figure 2: Illustration of the proposed Semi-supervised Anchor-Based Aggregation. The ignored pseudo label owing to low confidence is denoted as $-1$.

representation $\hat{q}_i = \theta_{glob}^{enc}(X_i)$, where $\theta_{glob}^{enc}$ refers to the global model's encoder applied to the client data $X_i$. Additionally, the pseudo label is generated as $\hat{y}_i = \theta_{glob}^{fc}(\hat{q}_i)$, where $\theta_{glob}^{fc}$ represents the global model's fully connected layer used for classification. Note that $q_i$ and $\hat{q}_i$ for each client are computed using **the same** $\theta_{glob}^{enc}$ and $\theta_{anch}^{enc}$, rendering **client data, $\mathcal{D}_{client}$ as the only variable**. We extract the features and compute the necessary pseudo-labels simultaneously, maintaining the computational complexity of $O(BE)$ per client per round, where $B$ represents the batch count and $E$ denotes the local epoch count.

At each round, we build a pseudo-aware running summation of the feature similarity between $q$ and $\hat{q}$ for pseudo-class $c$ with $M_c$ samples as follows:

$$\hat{w}_c = \frac{1}{M_c} \sum_{i=1}^{M_c} \frac{q_i \cdot \hat{q}_i}{\|q_i\| \cdot \|\hat{q}_i\|}, \tag{3}$$

where $q$ and $\hat{q}$ are normalized along the feature dimension so they lie on the unit sphere. Note $q$ represents the indexed features from the dictionary computed with the same initial weight distribution across all clients. Notably, the distance between $q$ and $\hat{q}$ can serve as a consistent measure for assessing divergence.

A high $\hat{w}_c$ indicates that the class representation of the client is close to random. Consequently, clients with lower $\hat{w}_c$ are more likely to approximate the global optimum. Given that all clients adhere to the same anchor and global model, a client with a lower $\hat{w}_c$ demonstrates greater diversity in its class representation. Therefore, $r_c = 1 - \hat{w}_c$ reflects the class importance of each client based on diversity measurement, which

should be normalized across clients as $\hat{r}_c = \frac{r_c}{\sum_K r_c}$. The final unlabeled client contribution can be computed by summing the distances to a scalar and the final averaged model, $\theta_{glob}$ is computed as:

$$\hat{r}_k = \sum_C \hat{r}_c \text{ , and } \theta_{global} = \sum_{k=1}^{K} \hat{\lambda}_1 \underbrace{\left(\frac{N_k^L}{N_{total}^L}\right)}_{\text{Supervised}} \theta_k + \hat{\lambda}_2 \underbrace{\left(\frac{\hat{r}_k}{\sum \hat{r}_k}\right)}_{\text{Unsupervised}} \theta_k, \tag{4}$$

where $C$ is the total number of classes. SemiAnAgg converges when unlabeled clients' pseudo labels become reliable. We discuss the convergence of SemiAnAgg in Section 5.3 and provide a detailed convergence analysis in Appendix B.

## 5 Experiments

### 5.1 Experimental Setup

**Datasets, Models, and Settings.** *We adhere to existing FedSemi benchmarks in all experimental settings*, as established by (Li et al., 2023) and (Liang et al., 2022), **while additionally expanding on multiple scenarios.** Specifically, we utilize four datasets to assess the effectiveness of our approach: SVHN, CIFAR-100 (Krizhevsky, 2009) (both the standard and its long-tailed imbalanced variant with an imbalance factor of 100), and the skin-lesion classification dataset, ISIC-18. Note that for imbalanced datasets, the imbalance is global, existing in both labeled and unlabeled data with class distribution mismatch between the label and unlabeled data. For all datasets, we reproduce the reported results of RSCFed (Liang et al., 2022) and CBAFed (Li et al., 2023) on the same non-IID federated partitioning publicly available, $Dir(\alpha) = 0.8$. While CBAFed utilize ResNet-18 ImageNet version detailed in (He et al., 2016) and RSCFed use a simple CNN, we find their lower-bound is not comprehensive. Thus, we reproduce their results by using the ResNet-18 CIFAR version detailed (He et al., 2016) for datasets with small spatial input dimensions (SVHN, CIFAR100), and the traditional ImageNet ResNet-18 in (He et al., 2016) for ISIC-18. This results in **improved reproduction of all baselines** than reported in CBAFed (Li et al., 2023).

To demonstrate the generalization performance, we evaluate the global model on the standard balanced test set for all datasets and report accuracy, following the conventions of previous FedSemi methods (Li et al., 2023; Liang et al., 2022). It should be noted that the ISIC-18 test set exhibits imbalanced class distribution, prompting us to provide a more comprehensive evaluation for this particular dataset.

**Implementation Details** (A detailed version in Appendix C.) To ensure a fair comparison, we maintain consistency in the training protocol, architecture, exact federated partitioning checkpoint, and other experimental settings across all methods. *The same warmup model is utilized for initialization in all reported tables unless otherwise stated*, which is trained for 250 rounds for SVHN, 250 rounds for ISIC-18, and 500 rounds for CIFAR-100. This is followed by FedSemi learning for an additional 500 rounds for SVHN, 500 rounds for ISIC-18, and 1000 rounds for CIFAR-100 until convergence. Notably, unlike RSCFed (Liang et al., 2022), and SemiAnAgg (ours), CBAFed (Li et al., 2023) benefit from temporal ensembling and a higher lower-bound. Consequently, we initialize CBAFed (Li et al., 2023) with a temporal ensembled model to maintain consistency.

**Baselines.** In our evaluation, we compare our results with state-of-the-art reproducible methods: RSCFed (Liang et al., 2022), IsoFed (Saha et al., 2023), and CBAFed (Li et al., 2023). To provide a more competitive baseline, we present in Appendix D and Appendix E additional baselines ablating local training strategies, specifically involving ACR (Wei & Gan, 2023) and SoftMatch (Chen et al., 2023) within the FedSemi framework.

### 5.2 Quantitative Comparisons with State-of-the-Art Methods

We strictly follow the experimental setting of the previous FedSemi benchmark (Li et al., 2023; Liang et al., 2022). This rigorous setting includes a single labeled client, which holds only 5% of the global dataset,

Table 1: Results on SVHN, CIFAR-100, CIFAR-100-LT and ISIC 2018 datasets under heterogeneous data partition. We report the results of all compared methods implemented by ourselves. Besides, the original results in papers are also reported and marked by †. We adhere to our baselines in the most rigorous setting (single labeled client), where one client contains a mere 5% of the global labels and 9 unlabeled clients. On imbalanced datasets, CIFAR-100LT and ISIC-18, we use logit adjustment (Ren et al., 2020) loss for all baseline methods.

| Labeling Strategy | Method | Dataset | | | |
|---|---|---|---|---|---|
| | | Balanced | | Imbalanced | |
| | | SVHN | CIFAR-100 | CIFAR-100-LT | ISIC-18 |
| Fully supervised | FedAvg (upper-bound) | 94.77 | 64.75 | 38.40 | 80.78 |
| | FedAvg (lower-bound) | 75.86 | 29.51 | 14.38 | 66.25 |
| Semi supervised | RSCFed† (Liang et al., 2022) | 76.74 | 28.46 | - | 67.21 |
| | CBAFed† (Li et al., 2023) | 88.07 | 30.18 | - | 68.29 |
| | RSCFed (Liang et al., 2022) | 81.84 | 31.98 | 15.13 | 69.85 |
| | IsoFed (Saha et al., 2023) | 82.48 | 32.49 | 16.98 | 68.49 |
| | CBAFed (Li et al., 2023) | 91.57 | 40.20 | 14.29 | 69.99 |
| | **SemiAnAgg (ours)** | **91.69** | **49.23** | **23.81** | **72.24** |

Table 2: Partially Labeled Clients.

| Method | SVHN | | ISIC-18 | |
|---|---|---|---|---|
| | Acc (%) | AUC (%) | B-Acc (%) | AUC (%) |
| RSCFed | 87.27 | 98.76 | 33.22 | 81.69 |
| CBAFed | 90.94 | 99.20 | 38.21 | 79.86 |
| **SemiAnAgg (ours)** | **92.57** | **99.50** | **42.62** | **88.34** |

Table 3: Effectiveness of aggregation methods.

| Agg. Methods | ISIC-18 Dataset | |
|---|---|---|
| | Acc. (%) | B-Acc. (%) |
| FedAvg (baseline) | 62.16 | 31.73 |
| FedAvg-Semi (our baseline) | 67.79 | 40.21 |
| **SemiAnAgg (ours)** | **72.24** | **48.77** |

alongside nine unlabeled clients. Additionally, we expand our experiments to include multiple scenarios and ablation studies. Table 1 reports the quantitative results on four benchmarks, including two balanced datasets SVHN and CIFAR-100, and two imbalanced datasets CIFAR-100-LT and ISIC-18 (skin-lesion).

**Results on the Balanced Global Setting.** As shown in Table 1, SemiAnAgg achieves the best accuracy on two balanced datasets, SVHN and CIFAR-100, outperforming the compared methods in terms of accuracy. Specifically, SemiAnAgg surpasses RSCFed, IsoFed, and CBAFed by 9.85%, 9.21%, and 0.12%, respectively, on SVHN, which is a relatively simple dataset. On a more challenging dataset, CIFAR-100, SemiAnAgg showcases a substantial improvement of 17.25%, 16.74%, and 9.03% respectively.

**Results with Imbalanced Global Setting.** Previous FedSemi methods did not consider the imbalanced global setting, which is critical in non-IID FedSemi, where class distribution mismatch between the label and unlabeled clients exists. To this end, we report the results on two imbalanced datasets (CIFAR-100LT and ISIC-18) while all baselines adopting logit-adjustments (Ren et al., 2020) to account for the class-imbalance. Our SemiAnAgg outperforms the SOTA baseline, CBAFed (Li et al., 2023) with 9.52% and 2.25% on CIFAR-100-LT and ISIC-18 respectively, achieving the best performance.

**Results with Partially Labeled Clients.** In Table 2, we compare the generic FedSemi approaches in the scenario of having partially labeled clients on the SVHN and ISIC dataset, a subset of the setting. In this partially labeled setting, all clients possess 10% of their data as labeled, while the remaining 90% is considered unlabeled. Our SemiAnAgg method demonstrates higher performance over CBAFed (Li et al., 2023), achieving a 1.63% increase in accuracy on the SVHN dataset and a 4.41% improvement in "B-Acc" on ISIC. Further experiments and insights are provided in Appendix C.

## 5.3 Ablation studies

**Effectiveness of SemiAnAgg.** The results of employing different aggregation strategies are presented in Table 3. Adopting FedAvg-Semi, which aligns with the principles of traditional semi-supervised optimization regimes, increases accuracy by 5.63% and "B-Acc" by 8.48% compared to the FedAvg baseline. FedAvg-Semi disentangles the aggregation of labeled and unlabeled clients, thereby avoiding the skewing of global

optimization towards unlabeled clients with potentially incorrect pseudo-labels. Unlike methods that assign weights based on the amount of data for unlabeled clients, our SemiAnAgg leverages data heterogeneity for valuation, substantially surpassing our simple baseline, FedAvg-Semi, by 4.45% on accuracy and 8.56% on "B-Acc".

**Effects of Random Anchor Initialization.** Figure 3 presents the results of our ablation study using different random seeds for the anchor model in SemiAnAgg compared to our simple baseline, FedAvg-Semi. The upper plot illustrates the performance on the balanced dataset, SVHN, while the lower plot shows results on the imbalanced dataset, ISIC-18. SemiAnAgg with different anchor seeds (denoted as $\theta_{anch}$ and $\theta_{anch,2}$) consistently outperforms FedAvg-Semi with consistent and faster convergence. Notably, the results demonstrate that the choice of anchor seed has a minimal impact on the final performance, indicating the stability of SemiAnAgg. It is worth mentioning that the random anchor model does not introduce additional information during local client optimization.

**SemiAnAgg Convergence.** We analyze the convergence behavior of SemiAnAgg on the SVHN dataset in Figure 4. Given that SVHN is a relatively simple task, the pseudo-labels become highly reliable ($\approx 99\%$) in Figure 4 (a). Consequently, SemiAnAgg **converges** to almost equal contribution from all clients ($1/9 \approx 0.111$) as shown in Figure 4 (b) . Notably, in the early rounds, SemiAnAgg values clients **9**, **7**, and **6** the most due to their respective distances from the anchor model. SemiAnAgg behavior is supported by the leave-one-out in Figure 4 (c) indicating these clients are the most significant (their removal results in the highest error rate). Conversely, clients **2**, **3**, and **8** are weighted less in SemiAnAgg, supported by the fact that their removal does not significantly impact performance (their removal results in the lowest error rate). To this end, SemiAnAgg's novel setup of using a randomly initialized anchor model can measure

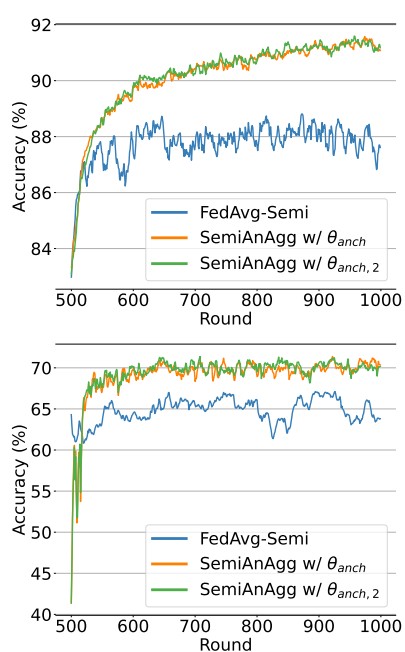

Figure 3: Ablation using FedAvg-Semi and SemiAnAgg with different random anchors. **Upper:** SVHN. **Lower:** ISIC-18.

not only the learned probably (most distant from the random anchor) but also consider the inter-client divergence (relative client distance). Unlike other FedSemi approaches (Liang et al., 2022; Li et al., 2023), SemiAnAgg's novel setup enables diverse unlabeled client aggregation.

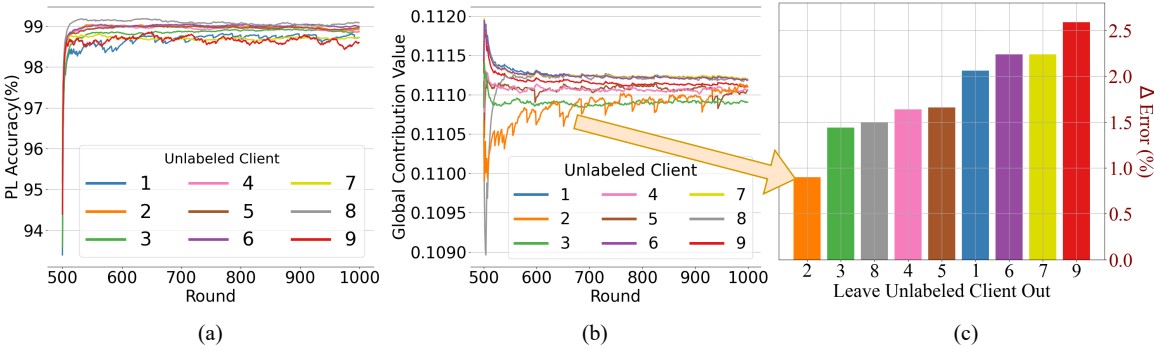

Figure 4: SemiAnAgg convergence analysis on the SVHN dataset. Note that as training progresses, the unlabeled client has reliable pseudo labels in (a) ($\approx 99\%$) given digit classification is relatively a simple task. SemiAnAgg converges to the clients contributing almost equally $1/9 \approx 0.1111$. (c) Client Importance by leaving one unlabeled client out (Each bar corresponds to the performance drop in FedSemi where an unlabeled client is removed).

Table 4: Ablation of varying numbers of clients and different $Dir(\alpha)$ on ISIC.

| | Number of Clients | | | | | | | |
|---|---|---|---|---|---|---|---|---|
| | 5 | | 10 | | 25 | | 50 | |
| | B-Acc (%) | AUC (%) | B-Acc (%) | AUC (%) | B-Acc (%) | AUC (%) | B-Acc (%) | AUC (%) |
| CBAFed | 52.58 | 86.45 | 41.12 | 86.04 | 33.64 | 81.03 | 28.76 | 74.08 |
| **SemiAnAgg (ours)** | **55.41** | **87.85** | **48.77** | **86.98** | **35.45** | **82.14** | **33.32** | **75.77** |
| | Dirichlet Parameters | | | | | | | |
| | Dir(0.1) | | Dir(0.5) | | Dir(0.8) | | Dir(2) | |
| | B-Acc (%) | AUC (%) | B-Acc (%) | AUC (%) | B-Acc (%) | AUC (%) | B-Acc (%) | AUC (%) |
| CBAFed | 33.12 | 60.23 | 41.05 | 85.01 | 41.12 | 86.04 | 45.83 | 87.68 |
| **SemiAnAgg (ours)** | **39.90** | **65.36** | **46.06** | **86.38** | **48.77** | **86.98** | **48.86** | **88.63** |

**Effects of Client Numbers.** Expanding the experimental scenarios, we consider a broader range of clients. Following the rigorous tests of (Liang et al., 2022; Li et al., 2023), we conduct an ablation study with 5, 10, 25, and 50 clients, where only one client is labeled and the others are unlabeled. Our SemiAnAgg outperforms the SOTA FedSemi approach, CBAFed (Li et al., 2023), with improvements of 4.56% in "B-Acc" and 1.69% in AUC on the ISIC dataset when federating 50 clients as shown in Table 4.

**Effects of Different Heterogeneous Levels.** Table 4 expands the ablations with scenarios of different heterogeneity and a broader number of unlabeled clients. Specifically, we examine the impact of client heterogeneity by employing a Dirichlet distribution $Dir(\alpha)$ with varying $\alpha$ values. A smaller $\alpha$ indicates greater heterogeneity. The results demonstrate that the SemiAnAgg method outperforms the state-of-the-art, CBAFed (Li et al., 2023), particularly under the most challenging conditions of heterogeneity with $\alpha = 0.1$, achieving improvements of 6.78% in "B-Acc" and 5.11% in AUC on the ISIC-18 dataset.

**Effects of Labeled Clients.** Figure 5 presents an ablation study on the impact of increasing the number of labeled clients on the ISIC dataset. Notably, all methods show improvements when the number of labeled clients is increased to two. Our SemiAnAgg method particularly outperforms the state-of-the-art CBAFed (Li et al., 2023) under the two-labeled-clients setting. Detailed metrics in Appendix C reveal a remarkable 12% increase in precision and a substantial 4.5% in "B-Acc".

**Privacy Implications of FedSemi.** While susceptibility to attacks within the FedSemi framework has not been studied previously due to pseudo-labels adding an extra layer of stochasticity, SemiAnAgg is more privacy-preserving than the SOTA FedSemi, CBAFed (Li et al., 2023), given the latter sharing the pseudo-class count of each unlabeled client, while SemiAnAgg opts for sharing pseudo-diversity scalars instead. Sharing pseudo-diversity scalars adds a layer of ambiguity against attempts to decipher the class distribution of unlabeled clients, stemming from the pseudo-diversity scalars being influenced not only by class count but also by the presence of minority attributes within a majority class, or conversely, a majority attribute within a minority class.

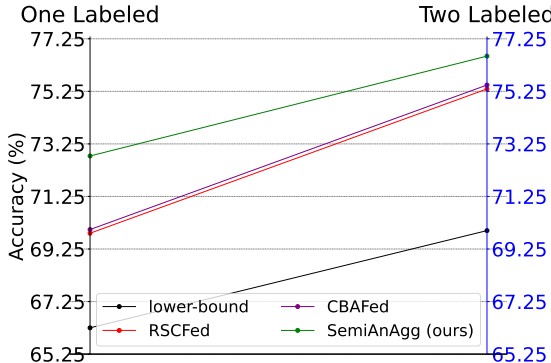

Figure 5: Performance improvement when the number of labeled clients is increased from one to two on ISIC.

**Additional Comments.** SemiAnAgg communication cost is the exact same as CBAFed (Li et al., 2023), 60% of RSCFed (Liang et al., 2022).

## 6 Conclusion

In this paper, we provide a new insight for FedSemi—highlighting the importance of measuring the divergence for unlabeled clients, which has been neglected in prior work. We introduce SemiAnAgg, a novel anchor-

based semi-supervised aggregation method that leverages a consistently initialized random anchor model across clients, allowing informative unlabeled clients to contribute more effectively during global aggregation. SemiAnAgg achieves new state-of-the-art results on four different benchmarks, offering unique insights for future research in FedSemi and semi-supervised imbalanced learning.

# 7 Limitations and Future Directions

In the absence of labeled data from unlabeled clients, SemiAnAgg approximates their contribution by comparing their data distribution through two consistently initialized models across clients. SemiAnAgg adds a layer of feature dictionary saving, nevertheless, it is negligible compared to the dataset size and the improvements obtained. Cosine similarity does not accurately represent the direction of the divergence given it shares only scalars. A more comprehensive approach, potentially involving multiple anchor models or feature representations, could better bound the similarity within the unit sphere. Nevertheless, sharing more independent features from clients might raise privacy concerns, which would need to be carefully addressed. Despite its simple design, *SemiAnAgg achieves state-of-the-art on four different benchmarks in FedSemi*, compared to state-of-the-art federated semi-supervised learning (FedSemi). FedSemi is widely addressed in classification, whereas extending it to imbalanced regression (Yang et al., 2021; Wang & Wang, 2023) remains a challenging future task. SemiAnAgg's lack of theoretical formulations is rather due to FedSemi complexity that has not been previously studied theoretically given the stochasticity of pseudo-labels highly dependent on initialization. We hope that our findings and insights inspire future approaches addressing the non-iid class mismatch and imbalanced problems in both centralized and decentralized settings.

# Broader Impact Statement

This paper introduces a novel federated learning aggregation, SemiAnAgg, whose goal is to enhance the utilization of unlabeled data across distributed networks, improving collaborative learning and ensuring data confidentiality. Uniquely, SemiAnAgg promotes the diversity of unlabeled clients, an aspect previously unexplored, by establishing a more equitable framework. While SemiAnAgg shares model weights and distance-derived scalars, these scalars do not reveal sensitive information due to their irreversible nature. We acknowledge the potential societal impacts, yet no specific issues must be highlighted here.

### Acknowledgments

This work was partially supported by grants from the National Natural Science Foundation of China (Grant No. 62306254) and the Hetao Shenzhen-Hong Kong Science and Technology Innovation Cooperation Zone (Project No. HZQB-KCZYB-2020083).

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

# Appendix for "Semi-supervised Anchor-Based Aggregation in Federated Learning"

The appendix is organized as follows:

- In Appendix A, we provide a detailed experimental analysis and discussion on federated self-supervised warmup, addressing the data dependency issues in FedSemi under limited data conditions.

- In Appendix B, we provide a more in-depth discussion of for the unlabeled client model aggregation (SemiAnAgg) and present empirical convergence analysis.

- In Appendix C, we discuss and present the detailed experimental settings as well as additional experiments. This appendix is organized as follows:

  - Appendix C.1: Comprehensive implementation details.
  - Appendix C.2: Information on dataset splitting and pre-processing.
  - Appendix C.3: Experiments involving more than one labeled client.
  - Appendix C.4: Experiments and discussion with partially labeled clients.
  - In Appendix D, we offer additional details on the integration of semi-supervised imbalance learning techniques, such as ACR (Wei & Gan, 2023), into the federated learning framework to establish a stronger baseline.
  - In Appendix E, we provide an ablation study that examines the impact of different local client's training strategies, including FlexMatch (Zhang et al., 2021) and SoftMatch (Chen et al., 2023).

## License of the assets

### License for the codes

We have reproduced the code for RSCFed (Liang et al., 2022) and CBAFEd (Li et al., 2023) and have achieved higher KPIs. The code for both implementations is publicly available. **We plan to make our code publicly available upon acceptance** licensed under the MIT License, which can be found at https://opensource.org/licenses/MIT.

### License for the dataset

For CIFAR-100 (Krizhevsky, 2009): We adhere to the terms provided by the CIFAR-100 dataset, which is released under the MIT License.

For SVHN (Netzer et al., 2011): We adhere to the usage of SVHN, which consists of Google Street View images, for non-commercial purposes.

For ISIC-18 (Tschandl et al., 2018): In accordance with the data use agreement, we comply with the attribution requirements of the Creative Commons Non-Commercial license (CC-BY-NC).

## A  Federated Self-Supervised Warmup

To ensure reliable pseudo-labels for unlabeled clients in initial stages, previous Federated Semi-Supervised Learning (FedSemi) approaches (Li et al., 2023; Liang et al., 2022) have employed a warm-up phase based on labeled clients using traditional FedAvg (McMahan et al., 2017). Unfortunately, as depicted in Figure 6, a poorly executed supervised warm-up can lead to model degradation, especially in the case of working with limited samples (Yang & Xu, 2020; Yan et al., 2022). To address this issue, intuitive federated self-supervised learning (Zhang et al., 2020; Lubana et al., 2022; Zhuang et al., 2022; Kim et al., 2023a) can be employed as a solution to alleviate the warm-up problem.

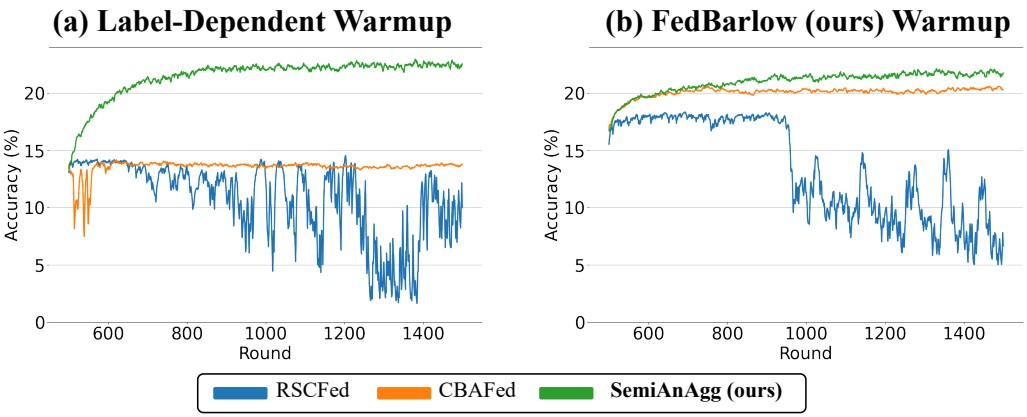

Figure 6: Comparative performance of RSCFed (Liang et al., 2022) and CBAFed (Li et al., 2023), and **SemiAnAgg (ours)** with (a) label-dependent warm-up and (b) self-supervised warm-up on CIFAR-100 LT.

In this paper, we **shed light** on the **impact** of an overlooked straightforward architecture, Barlow Twins (Zbontar et al., 2021), which is **not studied previously in the FedSelf framework**. We rename this architecture as **FedBarlow** through the minimization of Barlow-Twins redundancy objective locally, while the aggregation of local clients' models is performed using the FedAvg algorithm.

It is empirically demonstrated that FedBarlow, despite its simplicity, is a highly effective baseline in FedSelf.

In Table 5, we present comparative results on the SVHN and CIFAR-100 datasets in the context of self-supervised federated learning. We compare our approach, FedBarlow, with the clustering-based approach, Orchestra (Lubana et al., 2022). Our results demonstrate a significant improvement over Orchestra on CIFAR-100, achieving a 9.52% increase in

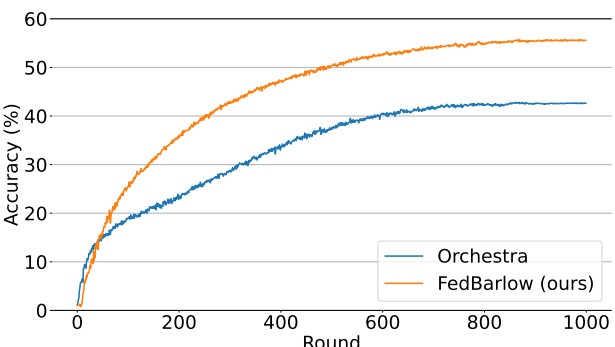

Figure 7: Comparison of Orchestra (Lubana et al., 2022) and FedBarlow (ours) in Federated Self-Supervised warmup on CIFAR-100. A federated online classifier is trained on the labels with a stop-gradient (SG).

linear evaluation performance. This improvement is even higher with 4.02% compared to the reported results in (Lubana et al., 2022) (55.89%). Furthermore, when training a federated online classifier using the 10 labeled clients, FedBarlow showcases remarkable improvements with a substantial increase of 14.6% and 10% on CIFAR-100 and SVHN, respectively. This highlights the importance of utilizing the Barlow Twin objective as a baseline in federated self-supervised learning, as it provides a consistent objective that promotes smooth alignment and leads to significant performance gains.

Table 5: Comparison of the accuracy between FedBarlow and Orchestra (Lubana et al., 2022) in a cross-silo 10-device setting on the CIFAR-100 and SVHN dataset. The upper bound corresponds to "Linear" probing over the dataset. while the lower bound pertains "Linear" on 10% of the data (one labeled client). FedCLR is an online classifier learned online between all clients with a stop gradient.

| Method | CIFAR-100 | | SVHN | |
|---|---|---|---|---|
| | Acc.(%) | AUC(%) | Acc.(%) | AUC(%) |
| Orchestra (Upper Bound) | 50.39 | 96.81 | 80.91 | 97.43 |
| **FedBarlow Ours (Upper Bound)** | **59.91** | **98.32** | **85.89** | **98.56** |
| Orchestra (Lower Bound) | 25.02 | 89.63 | 64.14 | 94.85 |
| **FedBarlow Ours (Lower Bound)** | **37.07** | **94.06** | **64.96** | **96.37** |
| Orchestra (FedCLR) | 42.81 | 95.49 | 69.03 | 95.08 |
| **FedBarlow Ours (FedCLR)** | **57.37** | **98.02** | **79.09** | **97.30** |

## B  SemiAnAgg Convergence

This section provides an in-depth discussion and empirical convergence analysis of the unlabeled client model aggregation approach, SemiAnAgg, on two datasets, SVHN and ISIC-18. Our SemiAnAgg focuses on the valuation of whether the model is learned properly while measuring divergence. SemiAnAgg introduces a novel client weighting strategy that aims to achieve a balanced contribution from unlabeled clients based on diversity measurements.

The SemiAnAgg weighting strategy offers an alternative metric to the computationally expensive impractical leave-one-out data valuation (Ghorbani & Zou, 2019). Unlike other approaches in semi-supervised learning that rely on metrics such as model confidence (Chen et al., 2023) or uncertainty ensembles (Chen et al., 2020), which have proven to be unreliable in imbalanced and class distribution mismatch scenarios, SSIL-CDM (as demonstrated in ACR (Wei & Gan, 2023)). SemiAnAgg takes a different perspective to address the challenges of federated non-iid semi-supervised training by considering diversity measurements as a criterion for client weighting. SemiAnAgg as a novel FedSemi adaptive weighting strategy shows competitive results to the baseline not using it as demonstrated in Figure 8.

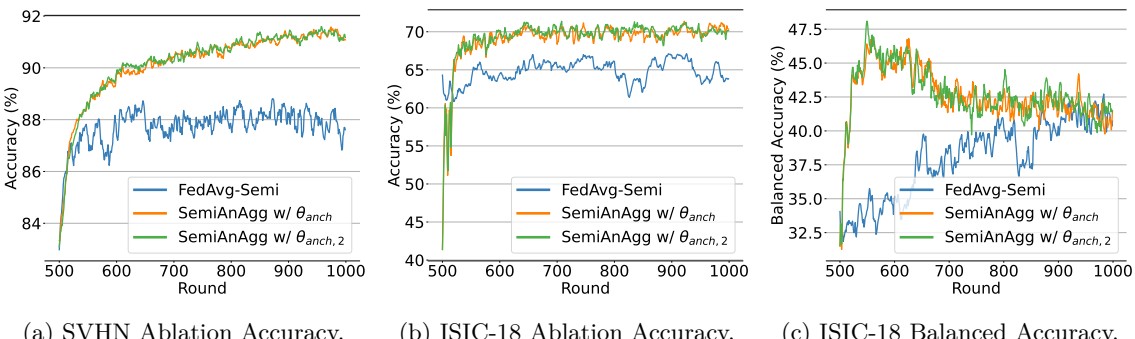

(a) SVHN Ablation Accuracy.  (b) ISIC-18 Ablation Accuracy.  (c) ISIC-18 Balanced Accuracy.

Figure 8: Ablation study comparing the use of SemiAnAgg and not using it on two different datasets, SVHN and ISIC-18. Note that ISIC-18 is highly imbalanced, so we also report the balanced accuracy.

On ISIC-18, an extremely imbalanced dataset, the global optimization can be skewed towards the majority classes, heavily biasing pseudo generation leading the model to get stuck at a local optimum. In Figure 9, we analyze the convergence of SemiAnAgg on ISIC-18. In Figure 9b, client 2 contributes the most to SemiAnAgg, which is supported by the leave-one-out analysis showing a significant error increase when client 2 is removed Figure 9c. Interestingly, client 4, despite having the highest pseudo-label accuracy in Figure 9a, contributes moderately in SemiAnAgg, as supported by its moderate ranking in the leave-one-out analysis. Contradictions observed: client 8 contributes substantially in SemiAnAgg, yet its removal does not result in a significant performance drop. This may be attributed to client 8 containing data with sensitive attributes (e.g., color, gender) that are not well represented in the imbalanced test set. Therefore, the leave-one-out analysis does not fully reflect its importance if such attributes are absent in the test set. Through these two

case studies, substantial improvements, and ablations, it becomes crucial to use our SemiAnAgg approach when training in federated semi-supervised learning settings.

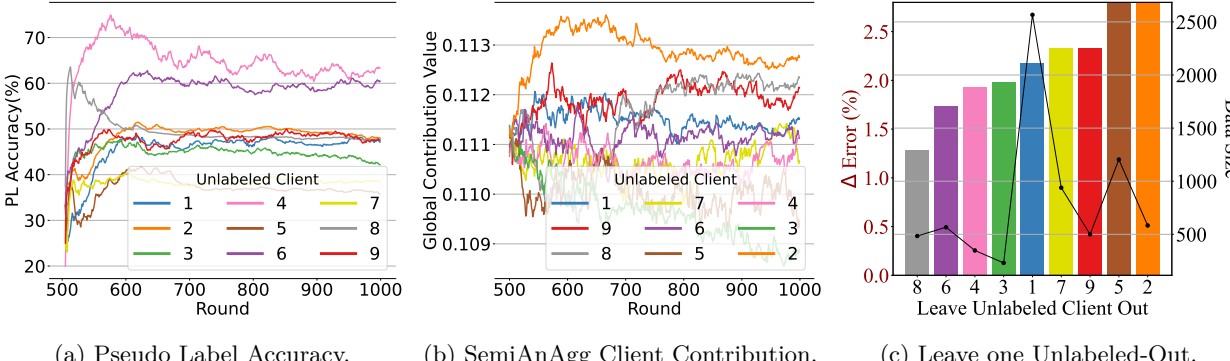

(a) Pseudo Label Accuracy.  (b) SemiAnAgg Client Contribution.  (c) Leave one Unlabeled-Out.

Figure 9: SemiAnAgg convergence analysis on the imbalanced ISIC-18. Note that unlike SVHN which is a relatively easy task. SemiAnAgg converges to a balanced version considering class proportion to avoid majority class skewing.

## C Additional Experiments

### C.1 Implementation Details

This section provides a detailed explanation of the implementation details for the main section (Sec. 4.1). We implement our method using the PyTorch framework (Paszke et al., 2019). In the Federated Semi-Supervised Learning (FedSemi) framework, SGD optimizer is used with a learning rate of 0.03 for labeled clients and 0.02 for unlabeled clients, following previous FedSemi approaches such as RSCFed (Liang et al., 2022) and CBAFed (Li et al., 2023). This difference in learning rates serves as an implicit regularization to prevent unlabeled clients from deteriorating the received global model in case of erroneous pseudo labels. A batch size of 64 is utilized for all datasets. The experiments were conducted on NVIDIA RTX 3090 GPUs.

The reported results in the main table follow a warm-up stage, similar to CBAFed (Li et al., 2023) and RSCFed (Liang et al., 2022). Specifically, CIFAR-100 and its long-tailed version are pre-trained for 1000 epochs, while the other datasets are pre-trained for 500 epochs. This is followed by an additional 1000 epochs for CIFAR-100 and its long-tailed version, and an additional 500 epochs for the other datasets. The optimal hyperparameter settings for other methods, RSCFed (Liang et al., 2022) and CBAFed (Li et al., 2023), as reported and presented in their respective code, are utilized. It is noted that, unlike other methods that involve scaling and extensive hyperparameters, *our SemiAnAgg does not contain any hyperparameters, and the parameters are solely dependent on the local learning method.*

For the local training procedure FlexMatch (Baseline) (Zhang et al., 2021), the default setting with the default confidence thresholds presented in TorchSSL, which is the official implementation of FlexMatch, is adopted. Better results are reproduced for all FedSemi baselines compared to the originally stated results in their tables (Li et al., 2023; Liang et al., 2022). The improvement for smaller spatial resolution datasets (SVHN, CIFAR) comes from utilizing an optimal architecture, CIFAR ResNet-18 (He et al., 2016), while for ISIC-18, the architecture (ImageNet ResNet-18 (He et al., 2016)) remains the same, and the improvement solely comes from utilizing RandAugment (Cubuk et al., 2020) as augmentation for all datasets. For all baselines, the classifier is learned without weight decay, while the backbone has a weight decay of 5e-4.

In experiments with self-supervised pre-training, the official Barlow Twins implementation is followed (Zbontar et al., 2021), but with some modifications. SGD optimizer is used with a learning rate of 0.008, and a scale loss of 0.01 is applied, which acts as an implicit scaling of the learning rate for biases and batch norm parameters. The choice of hyperparameters is based on the small batch size of 64 used for pre-training on all datasets.

## C.2 Dataset pre-processing

We explain in this section the steps taken for dataset splitting and pre-processing. The same dataset splitting for CIFAR-100, SVHN, and the skin dataset is adopted as in previous FedSemi approaches, RSCFed (Liang et al., 2022) and CBAFed (Li et al., 2023). It is ensured that the partition loading and splitting are derived from the same checkpoint for all evaluated methods, ensuring consistency.

Furthermore, the same pre-processing steps as those used in CBAFed (Li et al., 2023) and RSCFed (Liang et al., 2022) are adopted. These steps are likely described in more detail in the respective papers (Li et al., 2023; Liang et al., 2022). Additionally, a consistent data augmentation technique based on RandAugment (Cubuk et al., 2020) is employed. RandAugment (Cubuk et al., 2020) has been shown to improve performance for all baseline methods.

## C.3 More than One Labeled Client

In this section, we present the results of more than one labeled client setting for the ISIC-18 dataset. As expected, all methods show improved performance compared to the one labeled client setting.

In Table 6, we report the lower bounds for FedAvg (McMahan et al., 2017), which is used for the initialization of RSCFed (Liang et al., 2022) and our SemiAnAgg, and the lower bound of CBAFed (FedAvg with residual weight connection), which serves as the initialization for CBAFed (Li et al., 2023). The lower bound of CBAFed (Li et al., 2023) is significantly higher than that of FedAvg (McMahan et al., 2017), with an improvement of 4.9% in accuracy. However, while CBAFed (Li et al., 2023) exhibits enhanced performance in terms of accuracy (0.5% improvement) and AUC (3.0% improvement), it experiences a drop in balanced accuracy (1.8% decrease) compared to its lower bound. This observation is consistent with previous evaluations in the one-labeled client setting.

Table 6: Comparison of SemiAnAgg against CBAFed (Li et al., 2023) and RSCFed (Liang et al., 2022) on the ISIC-18 dataset. All methods are reported using logits adjustments (Ren et al., 2020) in labeled clients. Note that CBAFed benefits from an initialization with temporal ensembling while RSCFed and Ours are initialized with the same model without temporal ensembling.

| Labeling Strategy | Method | Client Num. | | Metrics | | | |
|---|---|---|---|---|---|---|---|
| | | labeled | unlabeled | Acc. (%) | AUC. (%) | Precision (%) | Recall (%) |
| Fully supervised | FedAvg (lower-bound) | 2 | 0 | 69.95 | 85.38 | 44.42 | 48.94 |
| | CBAFed (lower-bound) | 2 | 0 | 74.84 | 86.61 | 51.38 | 47.33 |
| Semi supervised | RSCFed | 2 | 8 | 75.34 | 89.67 | 58.89 | 49.10 |
| | CBAFed | 2 | 8 | 75.49 | 89.57 | 54.87 | 45.49 |
| | **SemiAnAgg (ours)** | **2** | **8** | **76.59** | **89.69** | **67.17** | **50.05** |

Remarkably, our SemiAnAgg achieves the best results across all four metrics, particularly demonstrating an impressive 12.3% increase in precision and a significant 4.5% improvement in recall (balanced accuracy) compared to state-of-the-art CBAFed (Li et al., 2023), despite being initialized from a worse starting point.

When compared to RSCFed (Liang et al., 2022), which maintains consistent initialization, our SemiAnAgg achieves greater improvements of 8.28% in precision and 0.95% in recall (balanced accuracy). It is important to note that RSCFed (Liang et al., 2022) removes noisy clients through average consensus, leading to the elimination of minority classes and consequently lowering precision (by predicting a high number of false positives, considering minority as majority) and recall (by predicting lower numbers for minority classes). In contrast, our SemiAnAgg addresses this issue by adaptively valuing unlabeled clients, implicitly assigning greater weight to clients with minority classes, thereby achieving a more balanced performance.

## C.4 Partially Labeled

In the context of FedSemi, we consider a scenario where clients have partial labeling, with each client having only 10% of its samples labeled. This setting presents a relatively easier local optimization compared to the case of one fully labeled client, but global optimization becomes challenging due to the non-iid data distribution. In this section, we present results for partially labeled clients on two datasets: SVHN and ISIC-18. The results are shown in Table 7.

Table 7: Comparison of SemiAnAgg against CBAFed (Li et al., 2023) and RSCFed (Liang et al., 2022) in the partially labeled setting on two datasets, SVHN and ISIC-18. For ISIC-18, all methods are reported using logits adjustments (Ren et al., 2020) based on the local labeled samples distribution. Note that CBAFed benefits from an initialization with temporal ensembling while RSCFed and SemiAnAgg (Ours) are initialized with the same model without temporal ensembling.

| Labeling Strategy | Method | Client Ratio | | Metrics | | | |
|---|---|---|---|---|---|---|---|
| | | labeled | unlabeled | Acc. (%) | AUC. (%) | Precision (%) | Recall (%) |
| | | | | Dataset 1: SVHN | | | |
| Fully supervised | FedAvg (upper-bound) | 100% | 0% | 94.76 | 99.67 | 93.95 | 94.82 |
| | FedAvg (lower-bound) | 10% | 0% | 83.80 | 98.51 | 82.10 | 85.15 |
| | CBAFed (lower-bound) | 10% | 0% | 84.15 | 98.45 | 82.60 | 84.83 |
| Semi supervised | RSCFed | 10% | 90% | 87.27 | 98.76 | 85.79 | 87.19 |
| | CBAFed | 10% | 90% | 90.94 | 99.20 | 90.43 | 90.18 |
| | **SemiAnAgg (ours)** | 10% | 90% | **92.57** | **99.50** | **92.02** | **92.28** |
| | | | | Dataset 2: ISIC-18 | | | |
| Fully supervised | FedAvg (upper-bound) | 100% | 0% | 80.78 | 93.53 | 61.55 | 69.22 |
| | FedAvg (lower-bound) | 10% | 0% | 66.95 | 82.17 | 35.39 | 41.62 |
| | CBAFed (lower-bound) | 10% | 0% | 68.30 | 82.26 | 36.62 | 40.78 |
| Semi supervised | RSCFed | 10% | 90% | 68.20 | 81.69 | 35.36 | 33.22 |
| | CBAFed | 10% | 90% | 67.40 | 79.86 | 37.00 | 38.21 |
| | **SemiAnAgg (ours)** | 10% | 90% | **70.39** | **88.34** | **40.71** | **42.62** |

(a) Accuracy on SVHN dataset.     (b) Area Under the Curve (AUC) on SVHN dataset.

Figure 10: Comparison with the state-of-the-art FedSemi on SVHN in the partial label setting.

We observed that the lower bound of CBAFed (Li et al., 2023) benefits from temporal ensembling, achieving a 0.4% and 1.4% improvement in accuracy on SVHN and ISIC-18, respectively, compared to the lower bound of FedAvg (McMahan et al., 2017). However, our SemiAnAgg surpasses CBAFed (Li et al., 2023) with a 1.6% improvement in accuracy on SVHN and a significant 2.9% improvement in accuracy and 4.4% improvement in balanced accuracy on the ISIC-18 dataset. Notably, the results for SVHN, which is a relatively easier dataset, are higher than those obtained in the one fully labeled client setting. This can be attributed to the dual environment learning in FL, which promotes invariant feature learning (Tang et al., 2022). On the other hand, for the highly imbalanced ISIC-18 dataset, it becomes challenging to learn the distribution in such a heterogeneous way, resulting in lower performance compared to the one fully labeled client setting. In Figure 10, we provide an analysis of the learning behavior of SemiAnAgg and the state-of-the-art CBAFed (Li et al., 2023) on the SVHN dataset in the partial label setting.

Comparing our SemiAnAgg to RSCFed (Liang et al., 2022), we outperform RSCFed (Liang et al., 2022) by 5.3% in accuracy on SVHN and by 2.1% in accuracy and a remarkable 9.4% in balanced accuracy on the ISIC-18 dataset. Additionally, SemiAnAgg achieves these improvements with less than 60% of the communication cost of RSCFed (Liang et al., 2022). It is worth noting that RSCFed (Liang et al., 2022) removes noisy clients through averaged consensus, which may exclude properly learned labeled clients that have unique classes not previously encountered. In contrast, SemiAnAgg incorporates bias learning for properly learned models in a balanced manner.

In the field of Federated Semi-Supervised Learning (FedSemi), previous approaches have often overlooked the aspect of semi-supervised imbalanced learning (SSIL) in their methods. Particularly in the non-iid setting, SSIL becomes apparent as depicted in Figure 11.

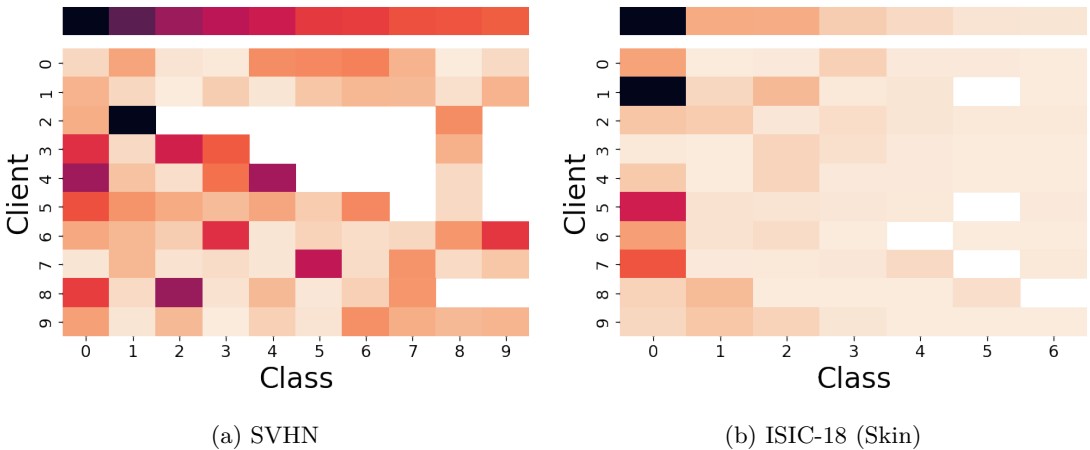

(a) SVHN                                   (b) ISIC-18 (Skin)

Figure 11: Clients Class distribution (Client 0 is labeled).

To establish a competitive baseline, we implemented the state-of-the-art SSIL method known as ACR (Wei & Gan, 2023). Our intuition is that ACR (Wei & Gan, 2023) could take into account both the imbalance resulting in the non-iid FL and the mismatch between labeled and unlabeled clients' class distributions. In the context of FL, we renamed this method as SSIL-CDM. Our previous experimental results have demonstrated that SSIL-CDM outperforms existing FedSemi methods, including RSCFed (Liang et al., 2022) and the state-of-the-art CBAFed (Li et al., 2023). However, it is worth highlighting that successful design strategies employed in FL play a crucial role in the ACR (Wei & Gan, 2023) method.

Table 8: Ablation of SemiAnAgg on the ISIC-18, skin. and the average per class accuracy B-Acc sensitive to imbalance which is the macro averaged recall in multi-class classification.

| | FedAvg-Semi | Adaptive Adj (Wei & Gan, 2023) | Dual Branch | Metrics | | | |
|---|---|---|---|---|---|---|---|
| | | | | Acc. (%) | AUC (%) | Pre (%) | B-Acc (%) |
| SSIL-CDM w/o FedAvg-Semi | ✗ | ✓ | ✓ | 66.60 | 82.47 | 50.33 | 35.88 |
| SSIL-CDM w/ FedAvg-Semi | ✓ | ✓ | ✓ | 71.34 | 85.12 | 46.36 | 43.14 |
| FedRoD | ✓ | ✗ | ✓ | 70.14 | **87.98** | 45.39 | 43.63 |
| FedRoD Dual | ✓ | ✗ | ✓ | 71.64 | 87.81 | 44.20 | 44.62 |
| **SemiAnAgg (ours)** | ✗ | ✗ | ✗ | **72.24** | 86.98 | **48.31** | **48.77** |

## D Integration of Semi-Supervised Imbalance Learning Techniques

The original ACR (Wei & Gan, 2023) requires a two-branch network with logit adjustment (Ren et al., 2020). Without prior knowledge about the global distribution (e.g., uniform, long-tailed, or imbalanced), one approach is to use labeled clients to adjust the logits of unlabeled clients. However, this can amplify an inaccurate distribution, leading to severe performance degradation (See orig-ACR FL in Figure 12). An improvement can be achieved by estimating a stable pseudo-label distribution with an Exponential Moving Average (EMA) locally using the global model. This modification namely SSIL-CDM achieves a reasonable performance in Figure 12. All models are initialized with FedAvg-Semi detailed in Equation 2.

FedAvg-Semi modifies the global optimization to be similar to local self-training optimization (Chen et al., 2023; Zhang et al., 2021; Sohn et al., 2020; Wang et al., 2023).

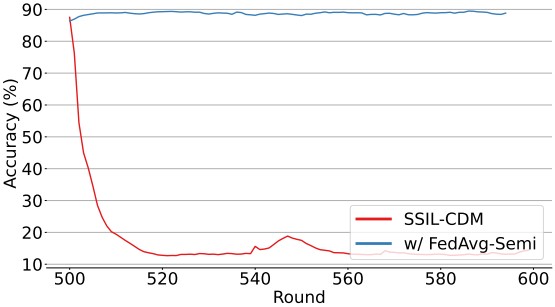

Figure 12: Comparison of ACR (Wei & Gan, 2023) with logit adjustment calculated from labeled clients (SSIL-CDM) and ACR with logit calculated locally with global model pseudo label distribution with EMA and FedAvg-Semi on SVHN dataset.

In Table 8, we present the results of different strategies on the ISIC-18 dataset. Using SSIL-CDM with FedAvg-Semi achieves improvements of 4.7% and 7.3% on accuracy and balanced accuracy compared to not using FedAvg-Semi (w/o FedAvg-Semi).

Our proposed approach, SemiAnAgg, outperforms our most competitive baselines with 0.9% and 5.6% on accuracy and balanced accuracy respectively. Notably, SemiAnAgg achieves this superior performance without requiring specifically tailored architecture design or the need for a dual branch architecture.

## E Client Local Training Methods

In this section, we present an ablation study to evaluate the impact of different local training methods, namely FlexMatch (Zhang et al., 2021) and SoftMatch (Chen et al., 2023), on the ISIC-18 dataset.

In Table 9, local semi-supervised learning methods show limited results due to the nature of FedAvg (McMahan et al., 2017), which does not fully account for the fact that labeled clients can provide more accurate information than unlabeled clients, regardless of data volume. While contributions such as CBAFed (Li et al., 2023) offer enhanced performance with a 9.41% increase in balanced accuracy, this still does not fully resolve the limitation.

Our FedAvg-Semi approach mitigates this limitation by balancing the empirical risk between labeled and unlabeled data, resulting in enhanced performance: an 8.48% increase in balanced accuracy for FlexMatch (Zhang et al., 2021) and an 8.57% increase in balanced accuracy for SoftMatch (Chen et al., 2023).

Table 9: Ablation study on local training methods for FL. † indicates methods specifically tailored for FL.

| Method | Acc. (%) | B-Acc (%) |
|---|---|---|
| FedAvg | | |
| CBAFed† (Li et al., 2023) | 69.99 | 41.12 |
| FlexMatch (Zhang et al., 2021) | 62.16 | 31.73 |
| SoftMatch (Chen et al., 2023) | 66.99 | 38.02 |
| FedAvg-Semi (our strong baseline) | | |
| FlexMatch (Zhang et al., 2021) | 67.79 | 40.21 |
| SoftMatch (Chen et al., 2023) | 72.39 | 46.59 |
| **SemiAnAgg (ours)** | | |
| FlexMatch (Zhang et al., 2021) | 72.24 | 48.77 |
| SoftMatch (Chen et al., 2023) | **72.59** | **49.84** |

Finally, by leveraging SemiAnAgg and considering the diversity of clients, we achieve the highest performance with a 7.65% increase in balanced accuracy for FlexMatch and 8.72% for SoftMatch compared with the state-of-the-art (SOTA) FedSemi approach, CBAFed (Li et al., 2023).

