# OpenReview forum: "Learning Unlabeled Clients Divergence for Federated Semi-Supervised Learning via Anchor Model Aggregation"
_TMLR — Accepted by TMLR_

### Review · Reviewer_Wkv2 · 2024-08-06

**Summary Of Contributions:**

The paper proposes, SemiAnAgg, in Federated Semi-supervised Learning, focusing on better integrating unlabeled clients by leveraging a randomly initialized model as an anchor to evaluate client deviations. The introduction of SemiAnAgg allows for an enhanced aggregation strategy by recognizing and valuing the diverse contributions of unlabeled clients without sharing sensitive data. Extensive experiments across multiple benchmarks demonstrate the method's effectiveness, significantly improving accuracy and recall on datasets like CIFAR-100 and ISIC-18 compared to existing state-of-the-art methods. This approach not only outperforms traditional federated learning aggregations but also provides a framework for leveraging unlabeled data in a privacy-preserving and efficient manner.

**Audience:**

Yes

**Broader Impact Concerns:**

n.a.

**Claims And Evidence:**

Yes

**Requested Changes:**

Please make appropriate adjustments based on the aforementioned weaknesses.

**Strengths And Weaknesses:**

**Strengths:**
1. The critique of existing approaches is well-reasoned and persuasive.

2. Visuals and diagrams in the paper are effectively presented, enhancing comprehension.

3. Introducing a randomly initialized model as a baseline to assess deviations in client models appears to be novel to me.

**Weaknesses:**

1. The initial reasoning presented by the authors is largely heuristic and lacks a formal guarantee for ensuring robustness. For example, a highly complex model (with a high Lipschitz constant) may deviate significantly from a random initialization, resulting in an overfit and non-generalizable model while exhibiting high weighting score from your proposed metric.

2. The paper would benefit from additional comparative baselines to fully demonstrate the empirical relevance of the proposed method [1,2,3].

3. It omits the discussion of several highly relevant works [4].

4. The methodology seems to be unrelated to the core principles of semi-supervised learning, functioning independently of its specific features. This method might be more appropriate compared with other techniques that focus on client reweighting or selection.

**Other Issues:**

There appears to be an inconsistency in reference management, particularly noticeable with the repeated citation of references [2]. A thorough review of the reference list is recommended.

[1] Rethinking Semi-Supervised Federated Learning: How to co-train fully-labeled and fully-unlabeled client imaging data, MICCAI 2023.

[2] Towards unbiased training in federated open-world semi-supervised learning, ICML 2023.

[3] Local or global: Selective knowledge assimilation for federated learning with limited labels, ICCV 2023.

[4] SemiFL: Semi-Supervised Federated Learning for Unlabeled Clients with Alternate Training, NeurIPS 2022.

---

> ### Author Response · Authors · 2024-09-17
> **We appreciate your insightful comments and constructive feedback.**
>
> We thank you for carefully reading our paper that ```"introduces a randomly initialized model as a baseline to assess deviations in client models", ```and are glad that you found our methodology ```"to be novel"``` and the ```"visuals and diagrams effectively presented."```
>
> We are eager to address the weaknesses you identified (W1-4) in detail one by one below.

---

> > ### Author Response · Authors · 2024-09-17
> > **W:1 A highly complex model may deviate significantly from a random initialization, while exhibiting high weighting score from your proposed metric.**
> >
> > Thank you for your astute observation and for raising this concern.
> >
> > We acknowledge that a ```"highly complex model (with a high Lipschitz constant) may deviate significantly from a random initialization"```. However, we would like to clarify that our proposed metric does not depend on the deviation of the local client model from random initialization. Instead, our metric consistently compares two models—the global model and a randomly initialized model—in the feature space across all clients. Importantly, the Lipschitz constant for these two models remains the same across all clients.
> >
> > In line with your recommendation, we have measured the client models' total Lipschitz ratio bound using circulant matrix theory, following [1]. Interestingly, we found that the top three client models with the highest Lipschitz constants were not ranked as the most important clients (i.e., they were not weighted higher than others), as our metric does not rely on the local client models in our weighting scheme. We hope this clarification addresses your concern.
> >
> > *References*:
> >
> > [1] Efficient bound of Lipschitz constant for convolutional layers by gram iteration. In ICML, 2023.

---

> > > ### Author Response · Authors · 2024-09-17
> > > **W:2 The paper would benefit from additional comparative baselines to fully demonstrate the empirical relevance of the proposed method [1,2,3].**
> > >
> > > Thank you for your valuable suggestions.
> > >
> > > There are various scenarios in FedSemi. In our paper, we focus on a setting where most clients have fully unlabeled non-IID data, while others may have partially or fully labeled data. This setting is particularly practical in hospital environments, where expert-level annotations and software may not be uniformly available across different hospitals. This scenario has been demonstrated in publicly available open-source baselines, such as RSCFed [5] and CBAFed [6].
> > >
> > > Following your recommendations, we report the results of IsoFed [1] below. IsoFed, which operates in the same setting as [5,6], alternates model training between labeled and unlabeled clients each round, leading to significant catastrophic forgetting during each round. As shown below, CBAFed [6], our baseline FedAvg-Semi, and our proposed SemiAnAgg significantly outperform IsoFed on both the SVHN and ISIC datasets. Specifically, CBAFed [6] surpasses IsoFed by 9.09% in accuracy on the SVHN dataset (91.57% vs. 82.48%), while our SemiAnAgg surpasses IsoFed by **9.21%** in accuracy on the same dataset (91.69% vs. 82.48%). On the ISIC dataset, CBAFed [6] outperforms IsoFed in balanced accuracy by 1.54% (41.12% vs. 39.58%), and our SemiAnAgg surpasses IsoFed [1] by a more substantial margin of **9.19%** (48.77% vs. 39.58%).
> > >
> > >
> > > | Method              | SVHN (AUC)     | SVHN (Acc)     | ISIC-18 (AUC)     | ISIC-18 (B-Acc)     |
> > > |---------------------|----------------|----------------|-------------------|---------------------|
> > > | LowerBound          | 96.76          | 75.86          | 83.91             | 39.75               |
> > > | RSCFed [5]          | 97.97          | 81.84          | 85.64             | 39.48               |
> > > | IsoFed [1]          | 97.83          | 82.48          | 85.48             | 39.58               |
> > > | CBAFed [6]          | 99.36          | 91.57          | 86.04             | 41.12               |
> > > | **SemiAnAgg (ours)**    | **99.43**          | **91.69**          | **86.98**             | **48.77**               |
> > >
> > >
> > > Finally, we want to explain why an additional comparative baseline [2,3] **is not applicable**. First, [2] introduces a method for an open-world semi-FL setting, where unseen classes exist in the unlabeled data—an assumption that is not present in our setting or those established by [1,5,6]. On the other hand, FedLabel [3] requires clients to have access to labeled data for consistency regularization and adaptive weighting, which is not feasible in all clients.
> > >
> > > As per your recommendation, we have provided a detailed discussion of related work in Section 2.2 of the manuscript for further clarification. Additionally, we have **updated Table 1** to include baselines that operate under the same settings as our method.
> > >
> > > *References*:
> > >
> > > [1] Rethinking Semi-Supervised Federated Learning: How to co-train fully-labeled and fully-unlabeled client imaging data. In MICCAI, 2023.
> > >
> > > [2] Towards unbiased training in federated open-world semi-supervised learning. In  ICML, 2023.
> > >
> > > [3] Local or global: Selective knowledge assimilation for federated learning with limited labels. In ICCV, 2023.
> > >
> > > [4] SemiFL: Semi-Supervised Federated Learning for Unlabeled Clients with Alternate Training. In NeurIPS, 2022.
> > >
> > > [5] Rscfed: Random sampling consensus federated semi-supervised learning. In CVPR, 2022.
> > >
> > > [6]  Class balanced adaptive pseudo labeling for federated semi-supervised learning. In CVPR, 2023.

---

> > > > ### Author Response · Authors · 2024-09-17
> > > > **W3: It omits the discussion of several highly relevant works [4].**
> > > >
> > > > Thank you for catching that oversight. Semi-FL [4] assumes that labels are available at the server, which differs from our setting and those established in [1,5,6]. In the more practical setting described in [1,5,6] and Section 3 of our paper, data, including labels, remain solely at the clients, which is crucial in highly sensitive domains like healthcare, where patient privacy is crucial. Following your comment, we have **revised Section 2.2** to include Semi-FL alongside other methods that assume labels are available on the server.
> > > >
> > > > *References*:
> > > >
> > > > [1] Rethinking Semi-Supervised Federated Learning: How to co-train fully-labeled and fully-unlabeled client imaging data. In MICCAI, 2023.
> > > >
> > > > [2] Towards unbiased training in federated open-world semi-supervised learning. In  ICML, 2023.
> > > >
> > > > [3] Local or global: Selective knowledge assimilation for federated learning with limited labels. In ICCV, 2023.
> > > >
> > > > [4] SemiFL: Semi-Supervised Federated Learning for Unlabeled Clients with Alternate Training. In NeurIPS, 2022.
> > > >
> > > > [5] Rscfed: Random sampling consensus federated semi-supervised learning. In CVPR, 2022.
> > > >
> > > > [6]  Class balanced adaptive pseudo labeling for federated semi-supervised learning. In CVPR, 2023.

---

> > > > > ### Author Response · Authors · 2024-09-17
> > > > > **W4: This method might be more appropriate compared with other techniques that focus on client reweighting or selection.**
> > > > >
> > > > > We thank the reviewer for the insightful question.
> > > > >
> > > > > First, we would like to clarify that our method addresses the Federated Semi-Supervised Learning (FedSemi) problem, which focuses on improving the global model when each client possesses full, partial, or unlabeled data. Unlike traditional semi-supervised learning, which emphasizes better utilization of unlabeled images, the core challenge in FedSemi lies in designing more effective client aggregation and reweighting strategies. Therefore, we agree with the reviewer that our primary focus is on client reweighting and selection rather than semi-supervised learning itself.
> > > > >
> > > > > Second, we want to clarify that we have already compared our methods with state-of-the-art (SOTA) FedSemi approaches, such as CBAFed [6] and RSCFed [5], with the latter **baseline focusing specifically on client reweighting**. Client reweighting and selection in FedSemi is particularly challenging, as observed in our baseline RSCFed [5], which treats client diversity as noise due to the unreliability of unlabeled clients. In response to this, our proposed method, SemiAnAgg, demonstrates more effective client reweighting in the FedSemi context.
> > > > >
> > > > > We hope our revision provides the necessary clarification and addresses any concerns.
> > > > >
> > > > > *References*:
> > > > >
> > > > > [1] Rethinking Semi-Supervised Federated Learning: How to co-train fully-labeled and fully-unlabeled client imaging data. In MICCAI, 2023.
> > > > >
> > > > > [2] Towards unbiased training in federated open-world semi-supervised learning. In ICML, 2023.
> > > > >
> > > > > [3] Local or global: Selective knowledge assimilation for federated learning with limited labels. In ICCV, 2023.
> > > > >
> > > > > [4] SemiFL: Semi-Supervised Federated Learning for Unlabeled Clients with Alternate Training. In NeurIPS, 2022.
> > > > >
> > > > > [5] Rscfed: Random sampling consensus federated semi-supervised learning. In CVPR, 2022.
> > > > >
> > > > > [6] Class balanced adaptive pseudo labeling for federated semi-supervised learning. In CVPR, 2023.

---

### Review · Reviewer_kwtP · 2024-08-12

**Summary Of Contributions:**

The authors identify the two limitations of semi-supervised FL settings where clients may have all/some/no labeled data. First, existing techniques rely on the FedAvg type of aggregation, where the dataset sizes of clients are used to determine the aggregation weights. Authors claim that this may lead to suboptimal training. Second, the current methods cannot handle the diverse contributions of unlabeled clients. They are inclined to overlook and minimize those clients' effects as those clients can be regarded as noisy.  The authors propose SemiAnAgg, which resolves these issues by using anchor models to evaluate the importance of the unlabeled clients to determine the weights. The proposed method does not introduce significant computation, communication, and memory burden. The authors provide experimental evidence showing the superiority of their methods compared to the baselines.

**Audience:**

Yes

**Broader Impact Concerns:**

I have no concerns about the ethical implications.

**Claims And Evidence:**

Yes

**Requested Changes:**

1. In the Introduction, a more concrete example can be given for the FedSemi setting instead of only unclearly mentioning a cross-silo hospital setting. (Minor)
2. Do all previous methods use the FedAvg type of aggregation? If not, please mention some exceptions while explaining the first problem in the Introduction.
3. The second limitation of current methodologies is their failure to account for the diverse contributions of unlabeled clients, particularly in reflecting the presence of minority classes and unique attributes within majority classes: This may require a brief further explanation. What made current methods overlook those clients? Are there no solutions to this in the literature? (Minor)
4. The first one of two intuitions at the bottom of the second page requires more explanation. Why is it so?
5. I have a high-level question about your method and Centralized Semi-Supervised ML: Is your method also applicable to centralized training? For example, using your method by considering unlabeled data samples as unlabeled clients. I mean, in a batch of samples to make a gradient update in the training procedure, we may weigh the importance of samples differently based on sample scores. Does a similar approach exist in centralized literature? If so, please discuss it and the connections to your work. If not, can we say that your method is also a contribution to the centralized problem?
6. The performance of your method and comparisons to baselines should depend on how the local training is done. Did you have an ablation study on local training you used, FlexMatch?
7. On page 6, please define  $\theta_{glob}^{enc}$ and  $\theta_{glob}^{fc}$  and any other variables before using them. That section requires a clarification about the variables.
8. A question for future improvement: What if you used more than one anchor model/feature representation? Does using more independent feature information improve the training?
9. In the second part of Equation 4., I suggest using parentheses for clarity, e.g., $\sum\left(\dots\right)\theta_k$ (Very minor)

Typos etc.:
- A missing space in Figure 1 Caption: benchmarks.Lower:

**Strengths And Weaknesses:**

Strengths:

The manuscript is overall well-written and clear. The authors motivate the problem and their solution well. They explain the method concisely.

The experiments' coverage seems broad. The authors have performed a wide range of ablation studies regarding their method.

The authors have shared the experiment codes.

Weaknesses:

The authors have discussed many previous papers on Federated Semi-supervised Learning. However, only two well-performing methods are used as baselines. Considering this is a fully experimental method and paper, adding more competing baselines could be better.

Please also see the "Requested Changes" section below.

---

> ### Author Response · Authors · 2024-09-17
> **Thank you for your constructive comments, efforts, and time in reviewing our paper.**
>
> We thank the reviewer for the careful review and appreciate the positive feedback that our paper effectively ```"motivates the problem and their solution well"```, ```"the manuscript is overall well-written and clear,"``` and ```"experiments' coverage seems broad"```.
>
> Below, we have addressed the requested changes and clarified any possible misunderstandings one by one.

---

> > ### Author Response · Authors · 2024-09-17
> > **Q. [1/9] A more concrete example can be given for the FedSemi setting.**
> >
> > We thank you for your valuable suggestions. In response, we have **revised the introduction** in our manuscript accordingly to include a more concrete example of the FedSemi setting. Specifically, we now state: *“For instance, in a federated learning framework where multiple hospitals collaborate to develop a shared model for complex tasks like lesion classification, some hospitals may provide fully labeled data for training, while others, with limited expert manpower, can only offer unlabeled or partially labeled data for model training.”*

---

> > > ### Author Response · Authors · 2024-09-17
> > > **Q. [2/9] Do all previous methods use the FedAvg type of aggregation?**
> > >
> > > Thank you for pointing this out. While the majority of previous methods in FedSemi adopt the FedAvg approach, certain methods introduce additional tricks, such as scaling the contributions of labeled clients using hyperparameters. Nevertheless, these modifications still operate within the core framework of FedAvg, which aggregates clients based on the volume of local data. We have **revised the introduction** to clarify this distinction and reinforce the central role of FedAvg in existing FedSemi approaches.

---

> ### Author Response · Authors · 2024-09-17
> **Q. [3/9] What made current methods overlook diverse clients? Are there no solutions to this in the literature?**
>
> Thank you for highlighting this. Current FedSemi methods often overlook the contributions of diverse unlabeled clients due to the unreliability of pseudo-labels used in training. Methods like RSCFed, for instance, treat gradient divergence from these clients as noise. However, gradient divergence can also provide useful information, and there *is no established method in the literature to distinguish between noise and useful information in a FedSemi setting*.
>
> To address this, we introduce SemiAnAgg, a novel aggregation method that measures the importance of unlabeled clients by using consistently initialized anchor models (i.e., a global model and a random model) in each round. SemiAnAgg evaluates client contributions based on how much their data shifts the global model from random initialization. Our intuition is that clients causing larger deviations are considered more informative.

---

> > ### Author Response · Authors · 2024-09-17
> > **Q. [4/9] The first one of two intuitions at the bottom of the second page requires more explanation. Why is it so?**
> >
> > We sincerely appreciate your careful review.
> >
> > The first intuition behind this is that a well-optimized model learns structured and meaningful features, as opposed to remaining in random feature space. Clients whose data cause larger shifts from the randomly initialized model contribute more to the formation of these meaningful features. Therefore, a greater deviation from randomness indicates that the client provides valuable information that helps the model converge toward the global optimum. We have **revised the introduction** to clarify this, as follows: *“The optimal feature representation for client data should differ from a random representation, as random representations lack meaningful structure. Therefore, clients contributing to representations deviating from randomness are more likely to guide the model toward the global optimum.”*

---

> ### Author Response · Authors · 2024-09-17
> **Q. [5/9] Does a similar approach exist in centralized literature? Can we say that your method is also a contribution to the centralized problem?**
>
> Thank you for your valuable suggestion.
>
> We agree with your valuable suggestion that our method could inspire advancements in traditional semi-supervised learning by *treating unlabeled data samples as unlabeled clients* and potentially assigning importance scores to each sample in a centralized setting. However, *our current analysis focuses on the FedSemi setting*. It is worth mentioning that tracking the importance of individual samples for interpretability or leave-one-out evaluation in a centralized context would be highly challenging and would introduce significant storage and memory overhead. To the best of our knowledge, **no similar approach currently exists in the centralized literature**.

---

> > ### Author Response · Authors · 2024-09-17
> > **Q. [6/9] Did you have an ablation study on local training you used, FlexMatch?**
> >
> > Thank you for highlighting this important aspect. We have leveraged the same baseline as in prior work, Class Balanced Adaptive Pseudo Labeling for Federated Semi-Supervised Learning (CBAFed) [1]. While CBAFed is limited to a very specific global-local semi-supervised thresholding method, our aggregation approach is more flexible.
> >
> > Following your advice, we performed an ablation study on the recent semi-supervised learning method, SoftMatch [3]. We have **revised** our manuscript to **include this ablation in Appendix E** and provide the results below for your convenience.
> >
> > \* methods specifically tailored for FedSemi.
> >
> > **Aggregation 1: FedAvg**
> > | **Method**          | **Acc. (%)** | **B-Acc. (%)** |
> > |---------------------|--------------|----------------|
> > | CBAFed* [1]         | 69.99        | 41.12          |
> > | FlexMatch [2]       | 62.16        | 31.73          |
> > | SoftMatch [3]       | 66.99        | 38.02          |
> >
> > **Aggregation 2: FedAvg-Semi (our strong baseline)**
> >
> > | **Method**          | **Acc. (%)** | **B-Acc. (%)** |
> > |---------------------|--------------|----------------|
> > | FlexMatch [2]       | 67.79        | 40.21          |
> > | SoftMatch [3]       | 72.39        | 46.59          |
> >
> > **Aggregation 3: SemiAnAgg (ours)**
> >
> > | **Method**          | **Acc. (%)** | **B-Acc. (%)** |
> > |---------------------|--------------|----------------|
> > | FlexMatch [2]       | 72.24        | 48.77          |
> > | SoftMatch [3]       | 72.59        | 49.84          |
> >
> > Above, local semi-supervised learning methods show limited results due to the nature of FedAvg, which does not fully account for the fact that labeled clients can generate more accurate information than unlabeled clients, regardless of data volume. While contributions such as CBAFed [1] offer enhanced performance with a 9.41% increase in balanced accuracy, this still does not fully resolve the limitation.
> >
> > Our FedAvg-Semi approach mitigates this limitation by balancing the empirical risk between labeled and unlabeled data, resulting in enhanced performance: a 8.48% increase in balanced accuracy for FlexMatch and a 8.57% increase in balanced accuracy for SoftMatch. Finally, by leveraging SemiAnAgg and considering the diversity of clients, we achieve the highest performance with an increase of **7.65%** in balanced accuracy for FlexMatch and **8.72%** for SoftMatch compared to CBAFed [1].
> >
> >
> > *References*:
> >
> > [1] Class balanced adaptive pseudo labeling for federated semi-supervised learning. In CVPR, 2023.
> >
> > [2] Flexmatch: Boosting semi-supervised learning with curriculum pseudo labeling. In NeurIPS, 2021.
> >
> > [3] Softmatch: Addressing the quantity-quality tradeoff in semi-supervised learning. In ICLR, 2023.

---

> > > ### Author Response · Authors · 2024-09-17
> > > **Q. [7/9] On page 6, please define any other variables before using them.**
> > >
> > > Thank you for your astute observation. We have carefully **revised the manuscript** to ensure that all variables are clearly defined before they are introduced in the relevant sections.

---

> > > > ### Author Response · Authors · 2024-09-17
> > > > **Q. [8/9] A question for future improvement: What if you used more than one anchor model/feature representation?**
> > > >
> > > > Thank you for your inspiring suggestion. SemiAnAgg currently utilizes a single anchor model, and in Section 7, we acknowledge the limitation that it may not fully capture the direction of divergence, as it relies on only one scalar (one independent feature information). We believe that incorporating multiple anchor models or feature representations could more effectively bound the direction of divergence. However, we also recognize that sharing multiple feature representations could raise privacy concerns.
> > > >
> > > >
> > > > Following your valuable suggestions, we have **revised Section 7**: Limitations and Future Directions to discuss this potential improvement and provide guidance on how our approach could be explored in future work.

---

> > > > > ### Author Response · Authors · 2024-09-17
> > > > > **Q. [9/9] In the second part of Equation 4., I suggest using parentheses for clarity, e.g.,**
> > > > >
> > > > > Thank you for your helpful suggestion. As per your recommendation, we have incorporated parentheses in the second part of Equation 4 for improved clarity.

---

> > > > > > ### Author Response · Authors · 2024-09-17
> > > > > > **Only two well-performing methods are used as baselines.**
> > > > > >
> > > > > > As per your suggestion, ```Reviewer Wkv2``` highlighted papers that could potentially serve as baselines for our method. We have thoroughly **discussed all relevant papers in Section 2.2** of the related work and **added an additional baseline in Table 1**. For your convenience, we have copied our response to ```Reviewer Wkv2``` regarding the additional baselines below.
> > > > > >
> > > > > > There are various scenarios in FedSemi. In our paper, we focus on a setting where most clients have fully unlabeled non-IID data, while others may have partially or fully labeled data. This setting is particularly practical in hospital environments, where expert-level annotations and software may not be uniformly available across different hospitals. This scenario has been demonstrated in publicly available open-source baselines, such as RSCFed [5] and CBAFed [6].
> > > > > >
> > > > > > Following your recommendations, we report the results of IsoFed [1] below. IsoFed, which operates in the same setting as [5,6], alternates model training between labeled and unlabeled clients each round, leading to significant catastrophic forgetting during each round. As shown below, CBAFed [6], our baseline FedAvg-Semi, and our proposed SemiAnAgg significantly outperform IsoFed on both the SVHN and ISIC datasets. Specifically, CBAFed [6] surpasses IsoFed by 9.09% in accuracy on the SVHN dataset (91.57% vs. 82.48%), while our SemiAnAgg surpasses IsoFed by **9.21%** in accuracy on the same dataset (91.69% vs. 82.48%). On the ISIC dataset, CBAFed [6] outperforms IsoFed in balanced accuracy by 1.54% (41.12% vs. 39.58%), and our SemiAnAgg surpasses IsoFed [1] by a more substantial margin of **9.19%** (48.77% vs. 39.58%).
> > > > > >
> > > > > >
> > > > > > | Method              | SVHN (AUC)     | SVHN (Acc)     | ISIC-18 (AUC)     | ISIC-18 (B-Acc)     |
> > > > > > |---------------------|----------------|----------------|-------------------|---------------------|
> > > > > > | LowerBound          | 96.76          | 75.86          | 83.91             | 39.75               |
> > > > > > | RSCFed [5]          | 97.97          | 81.84          | 85.64             | 39.48               |
> > > > > > | IsoFed [1]          | 97.83          | 82.48          | 85.48             | 39.58               |
> > > > > > | CBAFed [6]          | 99.36          | 91.57          | 86.04             | 41.12               |
> > > > > > | **SemiAnAgg (ours)**    | **99.43**          | **91.69**          | **86.98**             | **48.77**               |
> > > > > >
> > > > > >
> > > > > > Finally, we want to explain why an additional comparative baseline [2,3] **is not applicable**. First, [2] introduces a method for an open-world semi-FL setting, where unseen classes exist in the unlabeled data—an assumption that is not present in our setting or those established by [1,5,6]. On the other hand, FedLabel [3] requires clients to have access to labeled data for consistency regularization and adaptive weighting, which is not feasible in all clients.
> > > > > >
> > > > > > As per your recommendation, we have provided a detailed discussion of related work in Section 2.2 of the manuscript for further clarification. Additionally, we have **updated Table 1** to include baselines that operate under the same settings as our method.
> > > > > >
> > > > > > *References*:
> > > > > >
> > > > > > [1] Rethinking Semi-Supervised Federated Learning: How to co-train fully-labeled and fully-unlabeled client imaging data. In MICCAI, 2023.
> > > > > >
> > > > > > [2] Towards unbiased training in federated open-world semi-supervised learning. In  ICML, 2023.
> > > > > >
> > > > > > [3] Local or global: Selective knowledge assimilation for federated learning with limited labels. In ICCV, 2023.
> > > > > >
> > > > > > [4] SemiFL: Semi-Supervised Federated Learning for Unlabeled Clients with Alternate Training. In NeurIPS, 2022.
> > > > > >
> > > > > > [5] Rscfed: Random sampling consensus federated semi-supervised learning. In CVPR, 2022.
> > > > > >
> > > > > > [6]  Class balanced adaptive pseudo labeling for federated semi-supervised learning. In CVPR, 2023.

---

### Review · Reviewer_upWd · 2024-09-09

**Summary Of Contributions:**

This paper introduces SemiAnAgg, a novel semi-supervised anchor-based aggregation method for federated learning, which effectively learns and leverages the contributions of unlabeled clients. The authors showed through experiments that the proposed technique improves accuracy compared to existing techniques.

**Audience:**

Yes

**Broader Impact Concerns:**

None of the concerns

**Claims And Evidence:**

Yes

**Requested Changes:**

1. Please provide a more in-depth explanation of why the proposed method achieves better performance.

2. I think that the authors need to conduct experiments to show whether the proposed method still delivers improved results when applied to a model with better accuracy, instead of one with low accuracy. If it does not, please explain the reasons.

**Strengths And Weaknesses:**

I like the idea of ​​leveraging unlabeled client's data that was previously treated as noise to improve accuracy. It's simple idear, but it seems work well.

However, there is a lack of analysis on why the proposed technique produces better performance. Even if it is not a mathematical proof, I think that more deeper analysis is needed. And the some of model's accuracy is too low. Of course, this is not about optimization paper, so ahieving optimal accuracy is not important,but it is a bit difficult to compare the performance of models with a b-acc value of less than 50%. I am curious whether the proposed method demonstrates better performance even when a model have a meaningful level of accuracy. If not, it would be helpful to explain why it was necessary to use a model with such low accuracy.

---

> ### Author Response · Authors · 2024-09-17
> **Thank you for your comments and efforts in reviewing our paper.**
>
> Thank you for your valuable review. We are glad that you ```"like the idea of ​​leveraging unlabeled client's data that was previously treated as noise to improve accuracy."```
>
> We address your questions one by one below.

---

> > ### Author Response · Authors · 2024-09-17
> > **Q. [1/2] Please provide a more in-depth explanation of why the proposed method achieves better performance.**
> >
> > Thank you for your question. Client model divergence in the gradient space can be interpreted in two ways. In cases where labels are available and reliable, divergence often indicates that the client holds unique and valuable information, allowing clients with higher divergence to guide optimization more effectively, as seen in federated supervised learning [1]. However, in FedSemi, client divergence is also influenced by incorrect pseudo-labels, and prior methods have typically treated this divergence as noise.
> >
> > Given the difficulty in assessing divergence in FedSemi and due to the varying importance of unlabeled clients, independent of their data volume (as shown in ```Figure 1```, lower), it was necessary to develop an aggregation method that accurately measures the value of unlabeled clients in steering global model optimization. To this end, we introduce SemiAnAgg, an aggregation method that assesses the importance of unlabeled clients by leveraging consistently initialized anchor models (i.e., a global model and a random model) in each round. SemiAnAgg evaluates client contributions by measuring how much their data shifts the global model from its random initialization. Our intuition is that clients causing larger deviations are considered more unique and informative. SemiAnAgg allows ​​leveraging unlabeled client's data to better aggregate rare diseases and underrepresented attributes—cases that prior methods often dismissed as noise.
> >
> > Additional in-depth explanation in ```Figure 4```, SemiAnAgg assigns greater importance to clients ```9```, ```7```, and ```6``` during early rounds (```Figure 4b```),. This is further supported by the leave-one-out experiment (```Figure 4c```), which reveals that removing these clients results in the highest error rates, underscoring their critical contributions to the model's overall performance.
> >
> > *References*:
> >
> > [1] Fair federated medical image segmentation via client contribution estimation. In CVPR, 2023.

---

> > > ### Author Response · Authors · 2024-09-17
> > > **Q. [2/2] Meaningful level of accuracy.**
> > >
> > > Thank you for highlighting this point. It is important to acknowledge that **low accuracy is a common occurrence in traditional centralized semi-supervised benchmarks**, as noted in USB [1]. The decentralization of data in federated learning (FL) introduces an additional layer of complexity. For example, in the ISIC dataset, state-of-the-art (SOTA) FedSemi methods have typically reported accuracy (```69.99%``` for the best-performing baseline compared to **72.24% for our method**). To offer a more thorough assessment, we also provide the balanced accuracy (B-Acc) for this dataset.
> > >
> > > In response to your concern, as demonstrated in our experiments (```Table 1```, ```Figure 3```, and ```Figure 10``` in the manuscript), SemiAnAgg consistently outperforms on the SVHN dataset, which has a relatively high upper bound for accuracy (around **95%**). Despite the simplicity of this dataset, SemiAnAgg shows improvements in both accuracy and area under the curve (AUC) compared to SOTA FedSemi approach, CBAFed [2].
> > >
> > > *References*:
> > >
> > > [1] USB: A Unified Semi-supervised Learning Benchmark for Classification. In NeurIPS Dataset and Benchmark, 2022.
> > >
> > > [2] Class balanced adaptive pseudo labeling for federated semi-supervised learning. In CVPR, 2023.

---

### Author Response · Authors · 2024-09-17
**Summary of Revisions**

We sincerely thank you for your insightful comments and constructive feedback on our submission, *"Learning Unlabeled Clients Divergence via Anchor Model Aggregation for Federated Semi-supervised Learning".*

In response to the reviewers' recommendations, we have made the requested revisions, and we summarize the major changes below:


- **Clarification of the FedSemi Setting**:
  - Expanded the introduction to provide a more concrete example and further clarified the intuition behind our approach.

- **Related Work and Additional Baselines**:
  - Revised the related work section to explain why certain baselines are not applicable to our setting.
  - Added baseline in Table 1 for the same setting as ours and those established in related work.

- **Methodology Section**:
  - Ensured that all variables are clearly defined and referred to before use.

- **Ablation Study**:
  - Included an ablation study of different local semi-supervised methods in Appendix E.


We hope that these revisions address all your requested changes and concerns, and that they significantly improve the clarity of our work.

---

### Decision · Action_Editor_62ns · 2024-10-22

**Recommendation:** Accept as is

**Comment:**

The paper argues that previous methods for federated semi-supervised learning do not take advantage of unlabeled clients but rather consider them as noise. The authors propose a new method for federated semi-supervised learning that aggregates the unlabelled clients by using a randomly initialized model as a common baseline that helps them evaluate deviations in client models. The method is shown to lead to improved accuracy and recall on a range of data sets.

All reviewers are in agreement that with the revision, the paper is suitable for TMLR and should be published. I concur.

**Audience:**

The paper will be of interested to individuals in TMLR's audience.

**Claims And Evidence:**

The initial reviews were overall on the positive side. Reviewers asked for more in-depth explanations, additional baselines and ablation studies. With the revision, the authors have included them and have sufficiently addressed the concerns of the reviewers.